# Investigating the use of physics informed neural networks for dam-break scenarios

**Kinza Mumtaz** [1]*, **Muhammad Waasif Nadeem**[2], **Adnan Khan**[1], **Zahra Lakdawala**[3]

**1** Department of Mathematics, LUMS University, Lahore, Pakistan, **2** RPTU Kaiserslautern, Erwin-Schrödinger-Straße, Kaiserslautern, Germany, **3** Fraunhofer Institute for Wind Energy Systems, Küpkersweg, Oldenburg, Germany

* 19070011@lums.edu.pk

**Data availability statement:** All relevant data are within the manuscript and its Supporting information files.

## Abstract

The real-time forecasting of flood dynamics is a long-standing challenge traditionally addressed through numerical solutions of the Shallow Water Equations (SWEs). Numerical solutions of realistic flow problems using numerical schemes are often hindered by high computational costs, particularly due to the need for fine spatial and temporal discretization, complex boundary conditions, and the resolution of non-linearities inherent in the governing equations. In this study, we investigate the use of Physics-Informed Neural Networks (PINNs) to solve 1D and 2D SWEs in dam-break scenarios. The proposed PINN framework incorporates the governing partial differential equations along with the initial and boundary conditions directly within the training process of the network, ensuring physically consistent solutions. We conduct a systematic comparison of the solutions of SWE using the classical numerical scheme (Lax-Wendroff) with estimates of physics informed neural networks. For 1D SWE, a neural network is trained and validated on a dam-break problem, revealing that physics-informed models produce smoother but still acceptable estimates of wave propagation compared to standard numerical results. For 2D SWE, we consider various configurations of dam geometries along with varying initial profiles for water heights. Across all scenarios, reproduce the numerical baselines, albeit with limited accuracy, while avoiding spurious oscillations and numerical artifacts. Further tuning, achieved by incorporating numerical solutions into the PINN training, improved accuracy. This proof of concept demonstrates the potential of hybridized PINNs as a mesh-free, scalable, and generalizable framework for approximating solutions to nonlinear hyperbolic systems. Our results indicate that pre-trained, physics-informed models could serve as a viable alternative for real-time flood forecasting in complex domains.

## 1 Introduction

Hydrological models are critical tools for understanding and predicting the behavior of water systems across a variety of environmental applications. Traditionally, these models have relied on solving complex systems of partial differential equations (PDEs) through numerical simulation. While effective, such simulations can be computationally expensive and time

**Funding:** The author(s) received no specific funding for this work.

**Competing interests:** The authors have declared that no competing interests exist.

consuming, especially when dealing with large-scale models, multiple realizations, complex constraints, and varying parameter settings.

To address these limitations, machine learning frameworks have emerged as promising alternatives for predicting hydrological phenomena without relying solely on repeated numerical simulations. The objective of this study is to develop a model that improves simulation speed while maintaining the same accuracy of results. Among the many machine learning approaches, physics-informed machine learning offers a compelling solution by combining data-driven learning with physical constraints.

Previous research has explored a variety of methods for solving shallow water equations using classical numerical techniques. These approaches, while accurate, are often computationally intensive and difficult to scale. For instance, [1] employed a finite volume discretization to simulate flood scenarios and predict the time required for water to reach residential areas. Similarly, [2] proposed a hybrid finite-volume/finite-difference method for open channel flow, using a semi-implicit finite difference discretization for the dynamic-wave momentum equations, and a mass-conservative finite volume discretization for the continuity equation. These methods have demonstrated efficiency in modeling super-critical and trans-critical flows. In another study, [3] employed the MacCormack method to discretize the shallow-water equations for non-uniform flow in an inclined channel, allowing the simulation of the evolution of small amplitude monochromatic waves into roll waves. In the context of extreme hydrological events, [4] identified scientific approaches and challenges in mitigation water quality impacts, while [5] emphasized the importance of dam-break flood modeling due to the sudden onset, rapid expansion, and devastating consequences of such events. Predicting wave propagation during dam-break floods remains a critical challenge in hydrodynamic and hydrological engineering, where accurate simulation is vital for risk assessment and early warning systems.

Recent research has begun to explore the application of Physics-Informed Neural Networks (PINNs) in hydrology. For example, [6] described the convergence of machine learning and physics-based modeling, showing how their integration can lead to more robust and generalizable solutions. PINNs, introduced by [7–9], are trained not only on data but also on the governing laws of physics encoded as PDEs. This dual supervision enables PINNs to produce reliable predictions even in the presence of sparse or noisy data. The work of [10] examined the intersection of hydrology and machine learning, highlighting the need for scale-relevant theories in hydrology. They emphasize the need for hydrologists to clearly demonstrate where and when hydrological theory provides added value in simulation and forecasting. Physics-informed machine learning models, such as PINNs, provide a path forward by embedding physical laws directly into the learning process. Despite their promise, conventional PINNs face notable limitations. According to [9,11,12], PINNs still struggle with the simultaneous learning of differential equation solutions throughout the domain and at the boundaries, which often leads to optimization challenges and training inefficiencies.

The studies by [13,14] describe the weighted least-squares collocation approach used in PINNs, which combines physics-based and data-driven loss terms. While this approach enables the integration of physical laws into neural network training, it also comes with inherent limitations. These include the difficulty of accurately evaluating partial differential equation (PDE) residuals at initial and boundary conditions, the requirement for high regularity of the solution (i.e., smoothness and continuity), and the inability to naturally enforce conservation laws within the framework. Moreover, [15] highlighted two additional critical limitations that arise when applying PINNs to problems such as dam-break flood prediction. First, the accuracy of the solution is often compromised by the challenges of solving high-dimensional, non-convex optimization problems. These problems are susceptible to convergence to local

minima, which can hinder the network's ability to approximate complex dynamics accurately. The second limitation, not yet discussed here, further underscores the need for refined training strategies and architectural adjustments when deploying PINNs in real-world hydrodynamic scenarios. The second major limitation of PINNs involves their substantial computational cost and the prolonged training time required to integrate the governing partial differential equations (PDEs). As noted by Wang [16,17], a fundamental challenge in training PINNs lies in balancing two competing objectives: accurately learning the solution of the PDE within the domain and enforcing the correct behavior along the boundaries. This trade-off can hinder convergence and lead to suboptimal performance. [18] addressed the issue of gradient imbalances during training, which often result in inaccurate learning of the underlying PDE solutions. [19] identified several key drawbacks of PINNs, including poor scalability to real-world problems. Similarly, [20] emphasized that PINNs are inherently slower than conventional numerical solvers due to their reliance on gradient descent-based optimization methods. Moreover, highly deep networks are prone to vanishing gradient problems, making it difficult for the network to propagate learning signals through all layers effectively [19,21, 22]. This can cause the network to saddle at a local minima, impeding convergence. Finally, the fine-tuning of PINNs, such as choosing an optimal amount of data, network architecture, or loss weights, is typically done manually, which complicates their practical deployment and reduces reproducibility [9,23].

Despite these known limitations, there is a growing consensus that, when properly trained, neural networks can offer significant advantages in solving complex physical systems, especially in scenarios where data are sparse or incomplete. To this end, our work addresses the challenges of training PINNs by incorporating precursor simulations and evaluating a hybrid strategy designed to enhance stability and accuracy for dam-break flood predictions. In this study, we present a framework for solving 1D and 2D SWE using PINNs as an alternative to classical numerical simulations in dam-break flood scenarios. The proposed approach integrates data and physics based constraints to produce accurate predictions. Unlike purely data-driven models, PINNs do not depend exclusively on large datasets. Instead, they incorporate the governing physical laws, such as the SWEs, to constrain the solution space, allowing them to generate accurate predictions even with sparse, noisy, or incomplete data [9,13].

We train and compare three types of networks (data driven, physics-informed, and hybrid) against numerical reference solutions for a one-dimensional dam-break problem. Based on the insights from the 1D estimates, we extended the PINNs framework to 2D dam-break flood scenarios with varying initial water height configurations. In the 2D cases, PINNs struggled to provide good estimates when steep gradients were present. To address this, we hybridized PINNs with precursor numerical simulations during training, forming a hybrid model that resulted in more reliable predictions, particularly in regions with sharp discontinuities.

The paper is organized as follows: Sect 2 outlines the problem setup, including the mathematical formulation of the 1D and 2D SWEs, initial and boundary conditions, and the numerical method, specifically the Lax-Wendroff scheme, used to solve the equations. It also defines the dam-break scenarios in 1D and 2D, for which the numerical solutions serve as reference data to validate the outputs of the different neural network models described in Sect 3. Sect 4 presents the results, offering a comparative analysis of the performance of data-driven neural network (DDNN), physics informed neural network (PINN), and hybrid neural network (HNN) relative to one another and to the reference numerical solutions. Finally, Sect 5 summarizes the key findings, highlights current limitations, and outlines future directions for enhancing neural network models tailored to solving the SWEs.

## 2 Problem setup

In this section, we introduce one and two-dimensional shallow water equations, along with the Lax-Wendroff numerical scheme used to solve them. Numerical solutions are computed for the dam-break scenarios discussed in Sect 2.3. The solutions serve as reference benchmarks for assessing the accuracy of the neural network predictions.

### 2.1 Governing equations - Shallow water equations

The one-dimensional shallow water equation is obtained by considering a channel with a unit width and assuming negligible vertical velocity of the water. The horizontal velocity, denoted by $u(x,t)$, is assumed to be approximately constant across the channel cross-section. This assumption holds true for small waves that have a wavelength greater than the depth of the water. Additionally, the fluid is assumed to be incompressible. These equations are obtained by integrating the Navier-Stokes equations across the flow depth, under the assumptions of a hydrostatic pressure distribution, minimal channel slope, constant fluid density, and both free surface and bottom boundary conditions [24].

The hydrodynamics of a water channel can be described by the 1D shallow water equations [25] in vector form, which are written as:

$$\frac{\partial \mathbf{U}}{\partial t} + \frac{\partial \mathbf{F}(\mathbf{U})}{\partial x} = 0, \text{ where } \mathbf{U} = \begin{bmatrix} h \\ uh \end{bmatrix}, \text{ and } \mathbf{F}(\mathbf{U}) = \begin{bmatrix} uh \\ u^2h + \frac{1}{2}gh^2 \end{bmatrix} \tag{1}$$

Here, $h(t,x)$ is the water height (m), $u(t,x)$ is the water velocity (m/s), and $g = 9.8$ m/s$^2$ is the acceleration due to gravity. Similarly, the 2D shallow water equations are written as

$$\frac{\partial \mathbf{U}}{\partial t} + \frac{\partial \mathbf{F}(\mathbf{U})}{\partial x} + \frac{\partial \mathbf{G}(\mathbf{U})}{\partial y} = 0, \tag{2}$$

where

$$\mathbf{U} = \begin{bmatrix} h \\ uh \\ vh \end{bmatrix}, \mathbf{F}(\mathbf{U}) = \begin{bmatrix} uh \\ u^2h + \frac{1}{2}gh^2 \\ uvh \end{bmatrix}, \mathbf{G}(\mathbf{U}) = \begin{bmatrix} vh \\ uvh \\ v^2h + \frac{1}{2}gh^2 \end{bmatrix},$$

where the velocity components $u(t,x)$, $v(t,y)$ correspond to the x-and y directions, respectively. In these equations, $\mathbf{U}$ denotes the vector of conserved variables (e.g., water height and momentum components), while $\mathbf{F}$ and $\mathbf{G}$ represent the flux functions in the $x$– and $y$– directions, respectively. The main objective here is to determine the solution vector $\mathbf{U}$, which includes both water height and velocity as functions of time and space. The one-dimensional shallow water equation governs their evolution, offering insights into the dynamics and propagation of water waves in shallow channels.

The widely accepted method for simulating dam-break flood dynamics has traditionally been the use of the 2D SWEs. These equations have been extensively applied to model dam-break flows, and have been validated in numerous studies incorporating both experimental observations and field data, such as in [26]. As a simplified model representation of free-surface flows, the SWEs estimate water depth and horizontal momentum, forming a system of time-dependent, nonlinear PDEs expressed in conservative form with source terms.

## 2.2 Numerical scheme - Lax-Wendroff

In computational fluid dynamics, the Lax–Wendroff scheme is a finite difference method used to solve hyperbolic partial differential equations, such as the shallow water equations. In this study, we use it to simulate the dam-break problem, which models the rapid redistribution of water following the sudden removal or collapse of a dam, producing a sharp discontinuity in water height. We consider a rectangular computational domain $[0, L_x] \times [0, L_y]$ with time steps $t^n = n \Delta t, \quad n = 0, 1, 2, \dots$ and a uniform spatial grid

$$x_i = i \Delta x, \quad i = 0, \dots, N_x - 1, \qquad y_j = j \Delta y, \quad j = 0, \dots, N_y - 1,$$

At each time step, the scheme follows a two-step predictor-corrector procedure:

1. Predictor step: Estimate intermediate states at half a time step $t^{n+\frac{1}{2}}$ using the current solution $\mathbf{U}^n$ and the gradients of the fluxes.
2. Corrector step: Use the intermediate states to compute the fluxes and update the solution to the next full time step $\mathbf{U}^{n+1}$.

In one dimension, the scheme first averages the values at neighboring points and subtracts a correction term proportional to the flux difference. The updated solution is then obtained by applying a similar flux difference using the intermediate values:

$$\mathbf{U}_{i+\frac{1}{2}}^{n+\frac{1}{2}} = \frac{1}{2} \left( \mathbf{U}_i^n + \mathbf{U}_{i+1}^n \right) - \frac{\Delta t}{2\Delta x} \left( \mathbf{F}(\mathbf{U}_{i+1}^n) - \mathbf{F}(\mathbf{U}_i^n) \right),$$
$$\mathbf{U}_i^{n+1} = \mathbf{U}_i^n - \frac{\Delta t}{\Delta x} \left( \mathbf{F}(\mathbf{U}_{i+\frac{1}{2}}^{n+\frac{1}{2}}) - \mathbf{F}(\mathbf{U}_{i-\frac{1}{2}}^{n+\frac{1}{2}}) \right).$$

The same logic extends to two dimensions, where the intermediate states are computed in both $x$- and $y$-directions.

$$\mathbf{U}_{i+\frac{1}{2},j}^{n+\frac{1}{2}} = \frac{1}{2} \left( \mathbf{U}_{i,j}^n + \mathbf{U}_{i+1,j}^n \right) - \frac{\Delta t}{2\Delta x} \left( \mathbf{F}(\mathbf{U}_{i+1,j}^n) - \mathbf{F}(\mathbf{U}_{i,j}^n) \right),$$
$$\mathbf{U}_{i,j+\frac{1}{2}}^{n+\frac{1}{2}} = \frac{1}{2} \left( \mathbf{U}_{i,j}^n + \mathbf{U}_{i,j+1}^n \right) - \frac{\Delta t}{2\Delta y} \left( \mathbf{G}(\mathbf{U}_{i,j+1}^n) - \mathbf{G}(\mathbf{U}_{i,j}^n) \right),$$

The corrector step then updates the solution by accounting for flux differences in both directions.

$$\mathbf{U}_{i,j}^{n+1} = \mathbf{U}_{i,j}^n - \frac{\Delta t}{\Delta x} \left( \mathbf{F}(\mathbf{U}_{i+\frac{1}{2},j}^{n+\frac{1}{2}}) - \mathbf{F}(\mathbf{U}_{i-\frac{1}{2},j}^{n+\frac{1}{2}}) \right)$$
$$- \frac{\Delta t}{\Delta y} \left( \mathbf{G}(\mathbf{U}_{i,j+\frac{1}{2}}^{n+\frac{1}{2}}) - \mathbf{G}(\mathbf{U}_{i,j-\frac{1}{2}}^{n+\frac{1}{2}}) \right).$$

The Lax-Wendroff scheme achieves second-order accuracy in both space and time, offering better resolution of wave fronts and discontinuities compared to first-order methods. By computing numerical fluxes in each spatial direction based on local states, it enables stable and accurate simulations of transient hydraulic phenomena in both one and two dimensional settings.

## 2.3 Use cases for Dam-Break scenarios in 1D and 2D

In all cases, the dam is instantaneously removed at $t = 0$. The instantaneous collapse of the dam results in a shock wave propagating through the domain. The gravitational acceleration is set to $g = 9.81 m/s^2$.

**1D Dam-break scenario.** The idealized benchmark of a 1D dam break scenario, similar to the one described in [27], is considered. A vertical wall separates water heights in a channel of length $L = 1$ m, with the initial conditions described as follows:

$$h(x, 0) = \begin{cases} h_l, & 0 \leq x \leq 0.5, \\ h_r, & 0.5 < x \leq 1, \end{cases}$$

$$uh(x, 0) = 0,$$

Where $h_l = 1.0$ m and $h_r = 0.5$ m, denote the upstream and downstream height. Trivial zero-flux boundary conditions are applied.

The spatial domain has a length $L = 1$ m with a spatial resolution of $\Delta x = 0.005$ m, resulting in $N_x = 200$ spatial points. The simulation is performed over a time interval $t = [0, 0.4]$ s with a time step of $\Delta t = 0.001$ s, resulting in $N_t = 401$ time steps.

The numerical solution for the 1D-dam break scenario using the Lax-Wendroff scheme is shown in Figs 1 and 2. The numerical results will be used as a reference to validate the estimates of the neural network described in the next section.

**2D dam break.** We investigate four dam-break scenarios by varying the initial water height profiles, each corresponding to a distinct geometric configuration - stepped, rectangular, circular, and conical - as shown in Fig 3.

A brief description of the initial profiles for each variant is as follows:

- Variant 1 - Initial (sharp) Discontinuity in Water Height whereby the initial height of the water has a stepped profile with a sharp transition between two regions, leading to a distinct

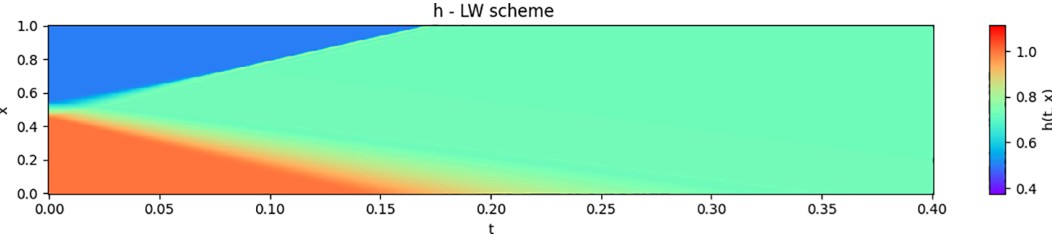

**Fig 1. Numerical solution for $h(t,x)$ using the Lax-Wendroff scheme.**

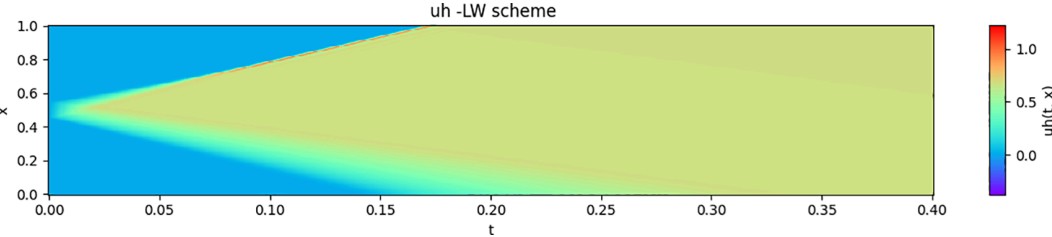

**Fig 2. Numerical solution for $uh(t,x)$ using the Lax-Wendroff scheme.**

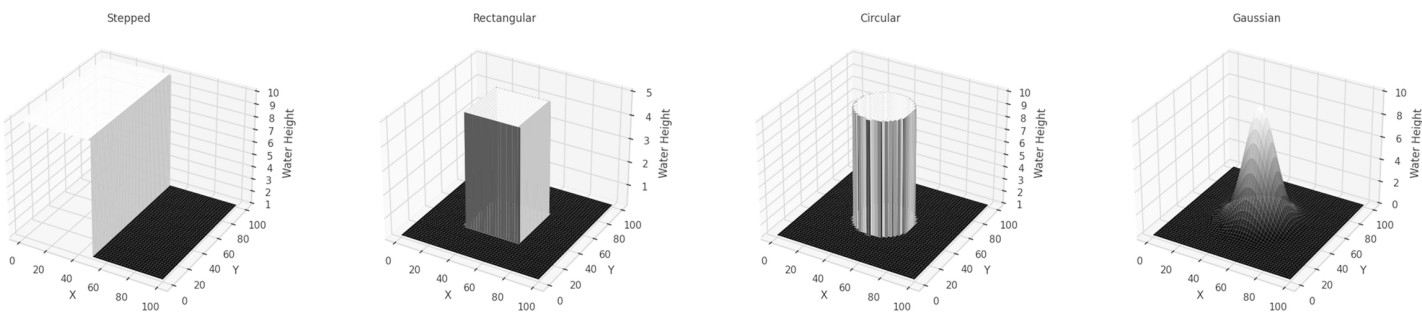

**Fig 3. Initial water height profiles for the four variants under consideration.**

redistribution of water once the dam gates are opened.

$$h(x,y,0) = \begin{cases} h_l = 10, & \text{for } 0 \leq x, y \leq 50 \\ h_r = 1.0, & \text{for } 50 \leq x, y \leq 100 \end{cases}$$

- Variant 2 - Rectangular shaped dam where the setup models a classical rectangular dam-break scenario, where a dam initially confines a higher water column within a specified rectangular region. The water height is set to 5.0 units inside this region and 0.2 units outside, creating a stepped initial profile. The dam barrier is assumed to be removed instantaneously, initiating a rapid redistribution of water. To simulate the undisturbed state before the dam break, the initial velocity is set to zero across the entire domain.

$$h(x,y,0) = \begin{cases} 5.0, & \text{if } 30.0 \leq x, y \leq 70.0, \\ 0.2, & \text{otherwise.} \end{cases}$$

$$u(x,y,0) = 0, \quad v(x,y,0) = 0.$$

- Variant 3 - 2D Circular Dam Break where the initial condition is a circular perturbation centered at $\left(\frac{L_x}{2}, \frac{L_y}{2}\right)$ with a radius of 20. Specifically, the function $h(x,y,0)$ is given by

$$h(x,y,0) = \begin{cases} 10.0, & \text{if } \sqrt{\left(x - \frac{L_x}{2}\right)^2 + \left(y - \frac{L_y}{2}\right)^2} \leq 20, \\ [1mm]1.0, & \text{otherwise.} \end{cases}$$

- Variant 4: 2D Gaussian Initial Profile, where the initial condition for the water height is modeled using a 2D Gaussian initial condition. This distribution defines the initial water height as follows:

$$h(x,y,0) = h_{\max} \exp\left(-\frac{(x - x_c)^2 + (y - y_c)^2}{2\sigma^2}\right)$$

where $h_{\max} = 10.0m$ is the maximum water height. The center of the Gaussian distribution is located at $(x_c, y_c) = (50.0, 50.0)m$, which represents the point of maximum water height. The standard deviation $\sigma = 10.0m$ determines how rapidly the water height decays as you move away from the center. It ensures a smooth transition of water height from the peak to the surrounding areas, which makes it particularly different from Variant 3.

The dam-break dynamics for each of the above initial water height profiles are simulated on a square domain of size $L_x \times L_y = 100 \times 100$ m$^2$ is considered. The spatial domain is discretized using a uniform Cartesian grid with $\Delta x, \Delta y = 0.2$ m, resulting in $N_x, N_y = 500$ points for $x$ and $y$ directions. The simulation runs for a total of $T = 1.0$ s, with timestep $\Delta t = 0.0001$ s.

Numerical solutions for Variants 1-4 using the 2-D Lax-Wendroff scheme are shown in Figs 4–7. Variant 1 features a stepped initial water height profile with a step size of 10 m, which leads to the formation of discontinuities in the wave front once the dam is removed. This results in more pronounced flow behavior, as shown by the numerical solution in Fig 4. This indicates a less controlled system response, where sharp transitions and disruptions are more pronounced. The numerical solutions for all variants are used as reference benchmarks for training and validating the neural network models presented in Sect 3. These solutions capture key flow features such as wave propagation and shock formation, serving as ground truth for evaluating the quality of PINN estimates.

## 3 Neural network

This section introduces a simple feed-forward convolutional neural network (FFCNN) used for training the networks for the problems outlined in Sect 2.

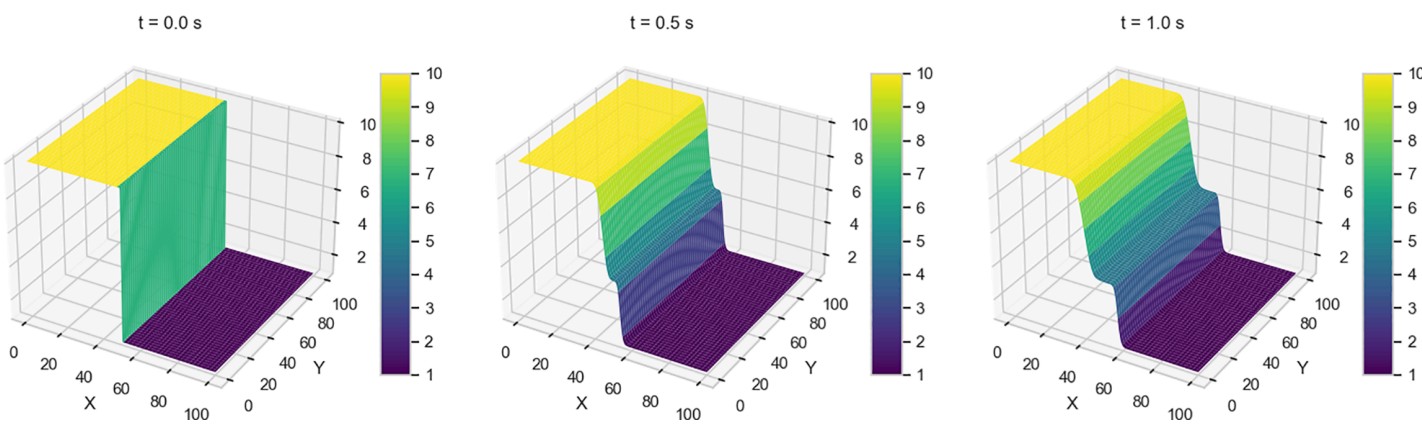

**Fig 4. Variant 1 - Water height using Lax Wendroff scheme at $t = 0, 0.5, 1$ s.**

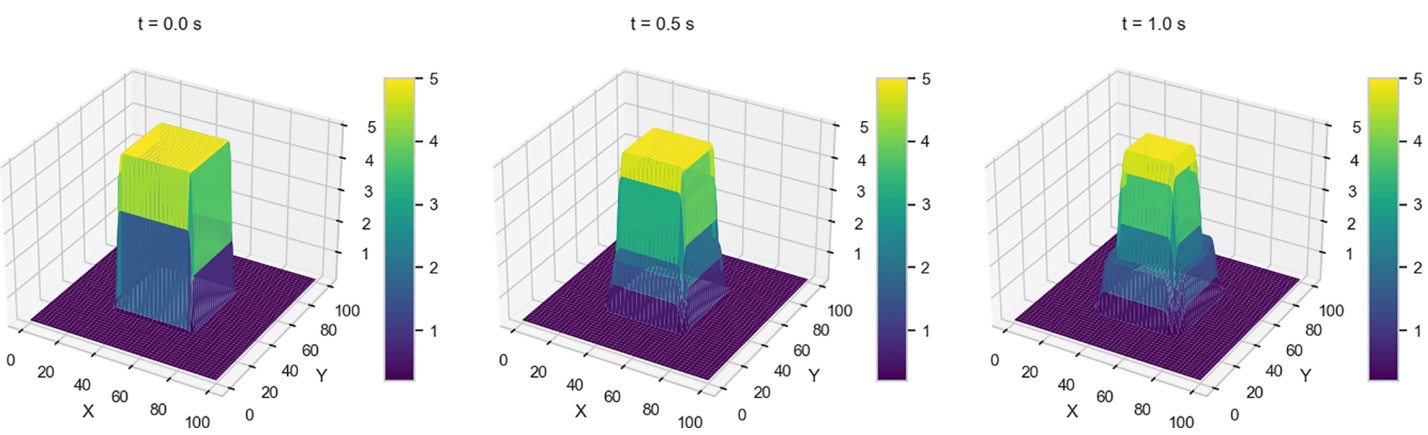

**Fig 5. Variant 2 - Water height using Lax Wendroff scheme at $t = 0, 0.5, 1$ s.**

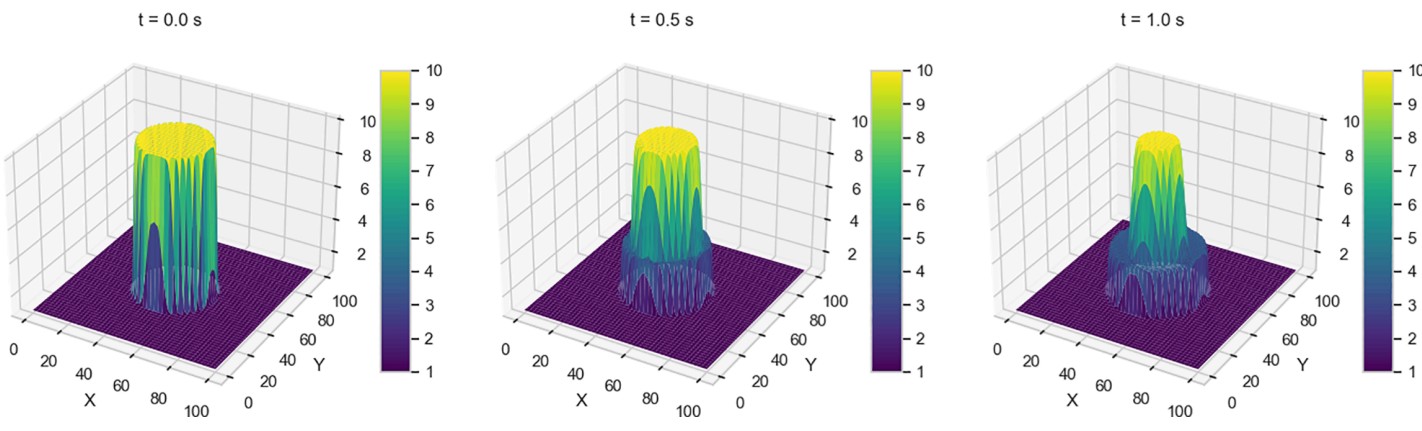

**Fig 6. Variant 3 - Water height using Lax Wendroff scheme at $t$ = 0, 0.5, 1 s.**

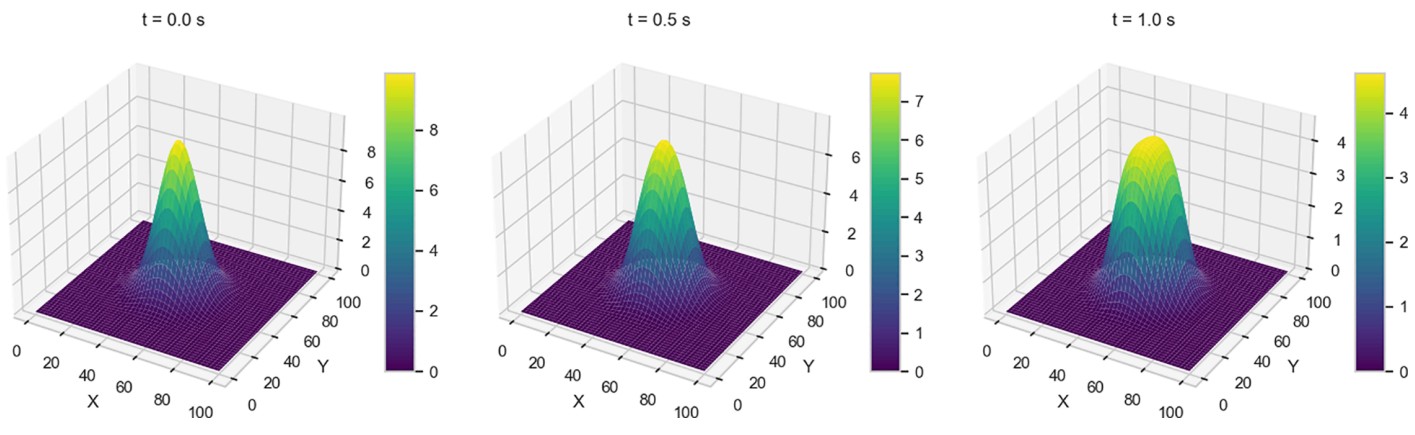

**Fig 7. Variant 4 - Water height using Lax Wendroff scheme at $t$ = 0, 0.5, 1 s.**

A neural network, denoted by $\mathcal{N}$, consists of an input layer, an output layer, and multiple hidden layers connected through weighted links and activation functions. With its $l$ hidden layers, the network performs a sequence of transformations to map input data to the desired output. Each neuron in a given layer is fully connected to all the neurons in the subsequent layer. These connections are characterized by weights $w^k$, biases $b^k$, and an activation function $\sigma$. The activation function introduces non-linearity into the model, enabling the network to learn complex patterns. The output of the $k$-th layer is a vector $u^k$:

$$u^k = \sigma\left(w^k u^{k-1} + b^k\right), \quad \text{for } k = 1, 2, \ldots, l-1. \quad \text{(Hidden layers)}$$

### 3.1 Data-driven neural network

A data-driven neural network is denoted as $\mathcal{N}(\mathbf{x}, \mathbf{t}, \mathbf{u}^*; \theta)$, where $\mathbf{t}, \mathbf{x}, \mathbf{u}^*(x, t)$ and $\theta$ represents the collocation points (i.e. spatial and temporal coordinates), the known values of the solution $\mathbf{u}^* = (h, uh)$, and the network parameters respectively. the input features are space-time coordinates (x,t) for 1D cases and (x,y,t) for 2D cases. The network parameters, $\theta = (w, b)$, are the set of trainable parameters (weights and biases) of the network. The network generates an approximate solution $\hat{\mathbf{u}} = (\hat{h}, \hat{u}\hat{h})$. The network $\mathcal{N}$ is trained using the input-output

pair $(\mathbf{x}, \mathbf{t})$ – $\mathbf{u}^*$ with the goal of learning to classify patterns between these pairs. To train a network using a backpropagation algorithm, its performance must be quantifiable and measurable. This is done by defining a loss function that evaluates how changes in the network parameters $\theta$ affect the output error, comparing the network's output to the desired target solution. Training of a neural network is thus a multi-objective optimization problem, where the loss function, typically expressed as mean squared error, needs to be minimized. A neural network requires an optimization routine to adjust its weights and biases in a way that minimizes the loss function defined on its output. This optimization is typically performed using a (stochastic) gradient descent algorithm or a quasi-Newton method. Training is considered complete when the network finds parameters that achieve the desired accuracy of the loss function.

In the case when the network is constrained to given data, the loss function is purely based on minimizing the discrepancy between the network prediction and the provided data. The L2-norm (mean squared error / MSE) loss function is a typical choice. This loss function measures the average squared difference between the predicted and target values across all data points.

$$\mathcal{L}_{\text{data}} = \mathcal{L}_{\text{data}}(\theta|(\mathbf{x}, \mathbf{t}, \mathbf{u}^*) = \frac{1}{N} \sum^{N} \|\hat{\mathbf{u}} - \mathbf{u}^*\|_2^2$$

In our case, the training data comes from the numerical solutions outlined in Sect 2, whereby $\mathbf{u}^* = (h^*, uh^*)$.

$$\mathcal{L}_{\text{data}} = \frac{1}{N} \sum^{N} \left[ \|\hat{h} - h^*\|_2^2 + \|\hat{uh} - uh^*\|_2^2 \right]$$

where $N$ is the number of collocation points.

## 3.2 Physics-informed neural network

In principle, a physics-informed neural network denoted as $\mathcal{N}(\mathbf{x}, \mathbf{t}, \mathbf{u}^*, physics; \theta)$ can be trained using provided rules/physics in addition to data (as described in Sect 3.1. The physics and data, in our case, are defined by a set of partial differential equations (PDEs) defined on a specific domain, along with boundary and initial conditions as described in Sect 2. The FFCNN for physics-informed training is similar to the one described for data-driven networks in Sect 3.1. However, because partial differential equations and boundary conditions may involve computing derivatives, the data-driven network is integrated with a gradient layer. This gradient layer takes the entire data-driven network as input and produces derivatives of the network's approximation to the solution of the partial differential equation. Fig 8 shows a data-driven neural network coupled with a gradient layer. The data-driven network consists of four hidden layers (each with 4 neurons) with a one-node input and a two-node output layer.

The goal of the training process is to adjust the network parameters so that the network accurately learns the solution to the given physical model for the given initial and boundary conditions. This involves calculating the derivatives of the output $\hat{\mathbf{u}}$ are then used to calculate the residuals of the partial differential equation. The norm of these residuals, along with the norms of residuals of the boundary conditions, contributes to the loss function. The partial differential equations are expressed in parameterized form, and the loss function is formally defined as follows:

$$\mathcal{L}_{\text{physics}} = \mathcal{L}_{\text{physics}}(\theta|(\mathbf{x}, \mathbf{t}, physics, \mathbf{u}^*) = \mathcal{L}_{PDE} + \mathcal{L}_{IC} + \mathcal{L}_{BC}$$

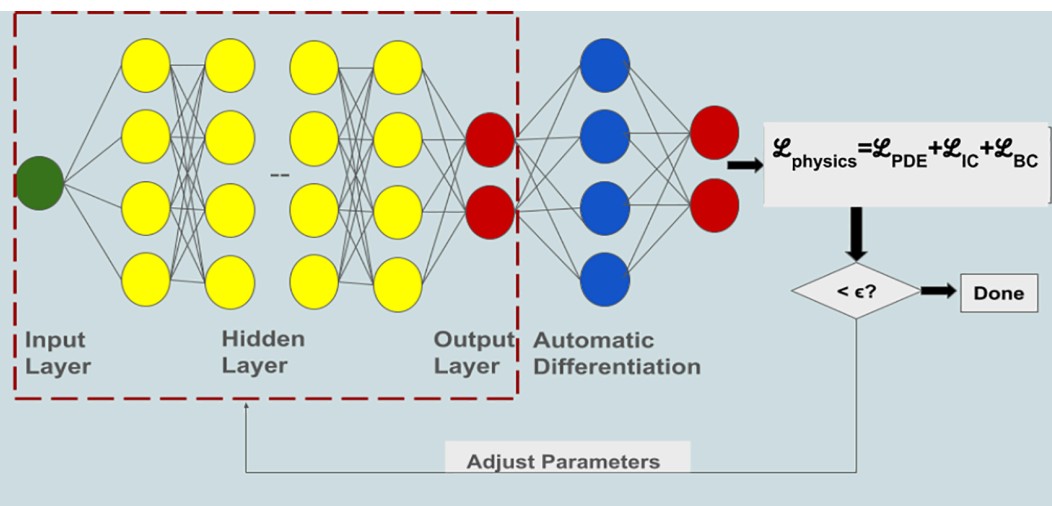

**Fig 8. Schematic Diagram of PINN architecture.**

The mean squared error (MSE) loss function is commonly used. For example, the physics loss for the shallow water equations can be written as:

$$\mathcal{L}_{\text{PDE}} = \frac{1}{N_{PDE}} \sum^{N_{PDE}} \left\| \frac{\partial \hat{h}}{\partial t} + \frac{\partial (\hat{u}\hat{h})}{\partial x} \right\|_2^2 + \left\| \frac{\partial (\hat{u}\hat{h})}{\partial t} + \frac{\partial}{\partial x} \left( \hat{u}^2 \hat{h} + \frac{1}{2} g\hat{h}^2 \right) \right\|_2^2,$$

Where $N_{PDE}$ is the number of collocation points in the interior of the domain. Similarly, the residuals for the initial and boundary conditions for the SWE can be written as

$$\mathcal{L}_{\text{IC}} = \frac{1}{N_{IC}} \sum^{N_{IC}} \left[ \left\| \hat{h}(x, 0; \theta) - h_0(x) \right\|_2^2 + \left\| \hat{u}\hat{h}(x, 0; \theta) - (uh)_0(x) \right\|_2^2 \right]$$

and

$$\mathcal{L}_{\text{BC}} = \frac{1}{N_{BC}} \sum^{N_{BC}} \left[ \left\| \hat{h}(0, t; \theta) - \hat{h}(1, t; \theta) \right\|_2^2 + \left\| \hat{u}\hat{h}(0, t; \theta) - \hat{u}\hat{h}(1, t; \theta) \right\|_2^2 \right].$$

Here $N_{IC}$ and $N_{BC}$ correspond to the number of collocation points for the initial and boundary conditions, respectively. The initial condition loss term quantifies the discrepancy between the predicted and true initial conditions, enabling the model to accurately capture the system's state at the initial time. Similarly, the boundary condition loss term measures the deviation from the prescribed boundary values, guiding the optimization of network parameters to satisfy the specified conditions across the spatial domain. Together with the PDE residual, these terms form a multi-objective loss function, where each component reflects a distinct physical constraint. The resulting optimization problem seeks a balance among these competing objectives, driving the model to simultaneously satisfy the governing equations, initial conditions, and boundary conditions as accurately as possible.

The residual terms associated with the PDEs and BCs typically involve the computation of derivatives with respect to space and time. These derivatives are efficiently computed using automatic differentiation, as implemented in machine learning frameworks such as PyTorch and TensorFlow.

### 3.3 Hybrid neural network

The network discussed in this section is derived through a hybrid strategy that combines data-driven and physics-informed strategies, as described in Sects 3.1 and 3.2. The network configuration remains as the one for PINN with the gradient layer.

The hybrid loss function integrates both the data-driven loss and the physics-based residual loss, with appropriate weighting over the spatiotemporal domain to guide the model toward solutions that respect both empirical data and the underlying physical laws described by a combination of PDE, IC, and BC.

$$\mathcal{L}_{hybrid}(\theta) = \mathcal{L}_{\text{physics}}(\theta|(\mathbf{x}, \mathbf{t}, physics, \mathbf{u}^*) = \mathcal{L}_{data} + \mathcal{L}_{\text{physics}}$$

In our use cases, the data component used in the hybrid loss function is derived from reference numerical simulations of the underlying physical system. Specifically, the solutions for $h(x,t)$ and $uh(x,t)$ obtained from numerical solvers are sampled at selected time instances across the spatial domain. These sampled values serve as additional constraints that guide the neural network during training. Even when using a limited number of such points, incorporating data helps steer the optimization toward physically meaningful solutions and can improve convergence behavior. The data-driven component is:

$$\mathcal{L}_{\text{data}} = \frac{1}{N_{data}} \sum_{(x,t) \in \mathcal{D}_{\text{data}}} \left[ \|\hat{h}(x,t;\theta) - h^{\text{num}}(x,t)\|_2^2 + \|\hat{uh}(x,t;\theta) - uh^{\text{num}}(x,t)\|_2^2 \right]$$

Here, $\mathcal{D}_{\text{data}}$ denotes the set of space-time points where numerical reference data is available, and $h^{\text{num}}(x,t)$ and $uh^{\text{num}}(x,t)$ represent the corresponding values from the numerical solution. The time values $t$ in this set correspond only to selected time instances rather than the full temporal domain. $N_{data}$ represents the total number of collocation points included in this data-driven term.

## 4 Results

In this section, the estimates of the neural network are validated against Lax-Wendroff numerical solutions, described in Sect 2, for the 1D and 2D dam-break scenarios.

### 4.1 1D dam-break

The estimates of three neural networks using the 1D SWE are presented here for the 1D dam break scenario described in Sect 2.3.

The DDNN was trained on input-output pairs $(t,x) \mapsto (h, uh)$, using a total of $N_x \times N_t = 200 \times 401 = 80200$ collocation points. At each of these points, the numerical solutions for water height $h$ and velocity density $uh$ were provided as training data.

The PINN consisted of two components: initial conditions and PDE residuals. For the initial condition, $N_{IC} = 200$ data points were randomly sampled at $t = 0$ over the spatial domain $x \in [0, 1]$. To enforce the underlying physics, $N_{PDE} = 200$ collocation points were randomly sampled within the spatiotemporal domain $x \in [0, 1]$ and $t \in [0, 0.4]$. In total, $N_{IC} + N_{PDE} = 400$ points were used to train the PINN.

The HNN combined three sources of training: numerical data, initial conditions, and PDE residuals. For the numerical component, simulation results were subsampled at discrete time instances $t = [0, 0.1, 0.2, 0.3, 0.4]$ across all spatial locations, resulting in $200 \times 5 = 1000$ training points. The training data for initial conditions and PDE residuals were selected similarly to the PINN but with 1000 points each. Altogether, the HNN was trained on 3000 points.

The network architecture consists of 10 hidden layers with 30 neurons each, using the *tanh* activation function. The training is performed using the LBFGS optimizer. This architecture was inspired by the hybrid training framework introduced by [13].

The estimates for *h* and *uh* from the different neural networks are compared in Figs 9 and 10 against the reference numerical solution. The results from the Lax-Wendroff scheme have a 'non-physical' oscillatory behavior due to the sharp gradients. In contrast to numerical estimates, the neural network-based estimates for *h* and *uh* are smoother, even in regions around shocks. The relative errors, shown as a histogram in Fig 11, indicate that the DDNN achieves high accuracy - within 2% relative error - while also avoiding the spurious oscillations typically observed in numerical schemes. PINN produces estimates within a 5% relative error range. However, the highest errors are a slight underestimation of *h* and *uh* at locations of sharp gradients, which may be attributed to the model's limited capacity to resolve discontinuities or potential overfitting in these regions. The HNN also performs within the 5% for *h* but a much larger relative error of up to 25% for *uh*, exhibiting a higher frequency of larger errors compared to both the DDNN and PINN. These results underscore the trade-offs between different neural modeling strategies in terms of accuracy and robustness near discontinuities. The PINN solution shows the smallest error throughout the domain, while HNNs and DDNNs display higher variance and larger deviation from the reference solution, particularly in high-gradient regions.

The relatively high error observed in the HNN predictions may stem from the model's difficulty in effectively balancing data-driven and physics-based losses. Although the HNN

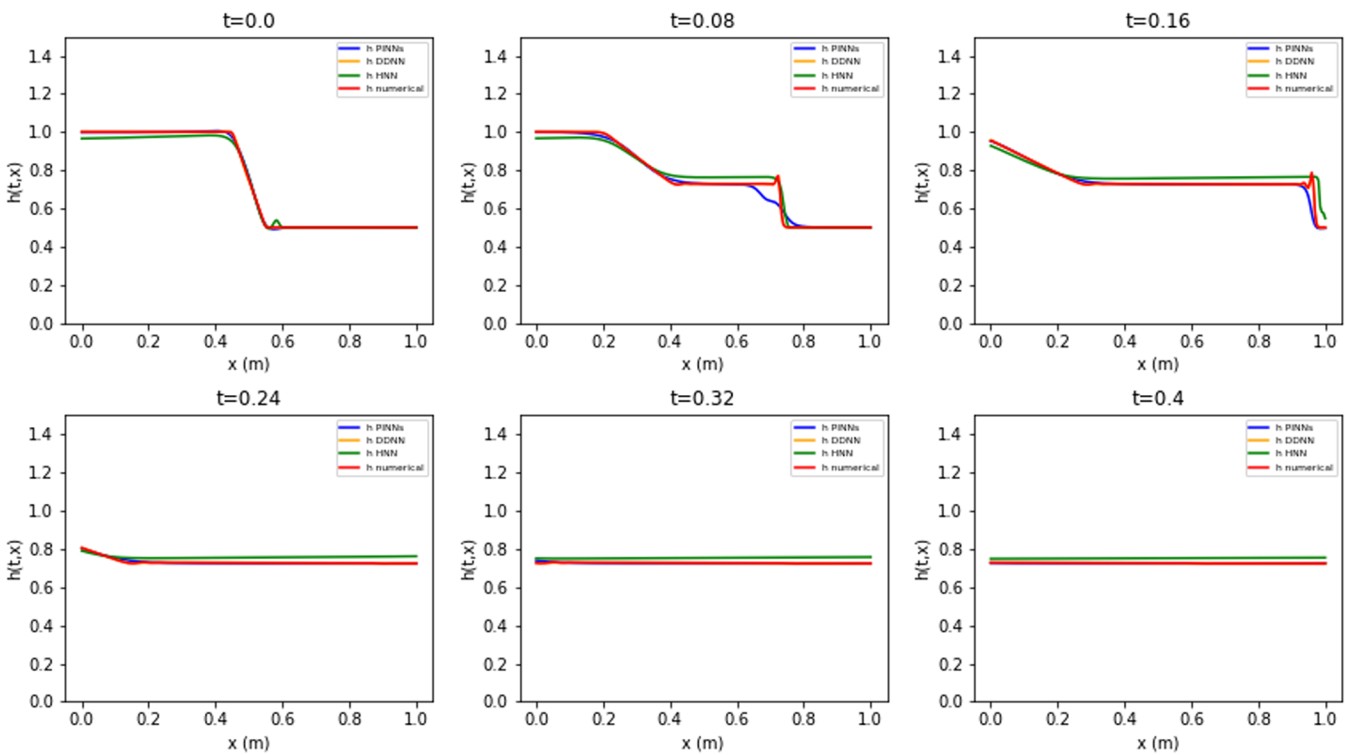

**Fig 9. Estimates of PINN, HNN, and DDNN for *h(x,t)* compared to reference Lax-Wendroff solution.**

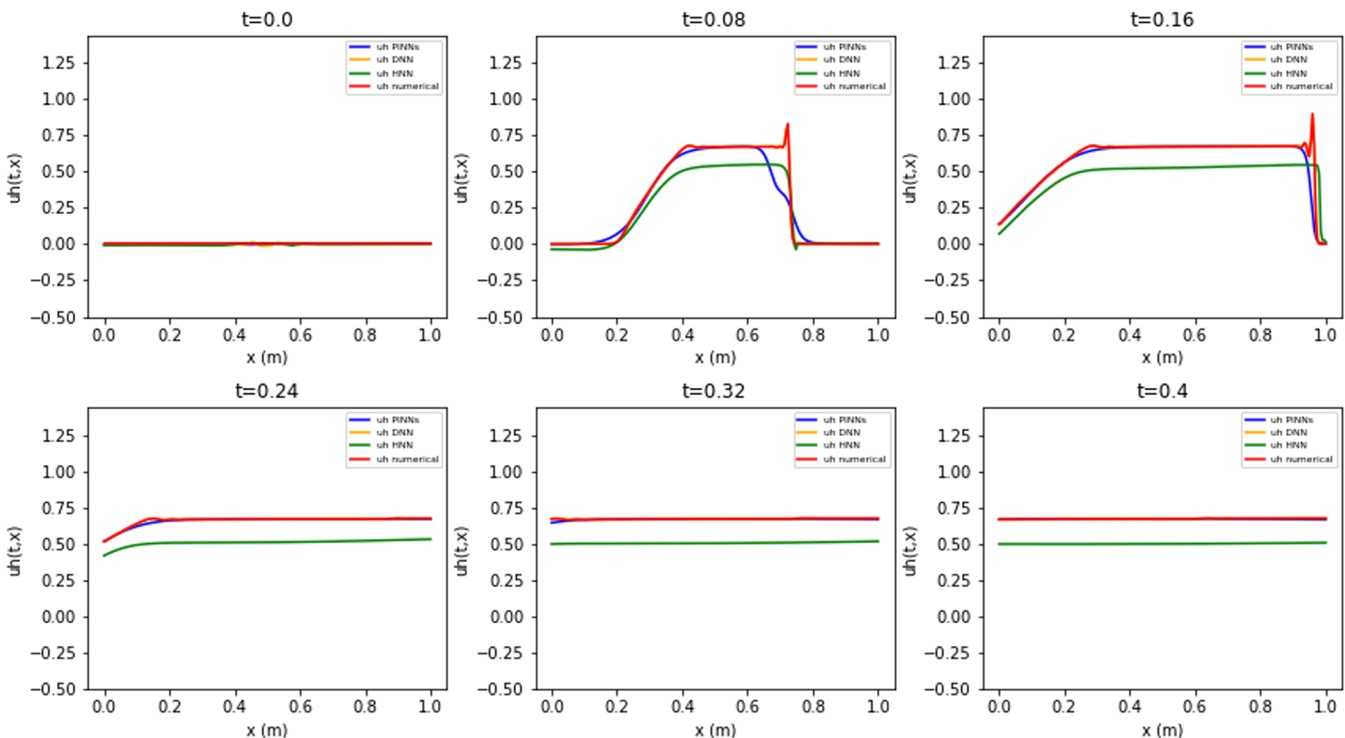

**Fig 10. Estimates of PINN, HNN, and DDNN for *uh*(*x*,*t*) compared to reference Lax-Wendroff solution.**

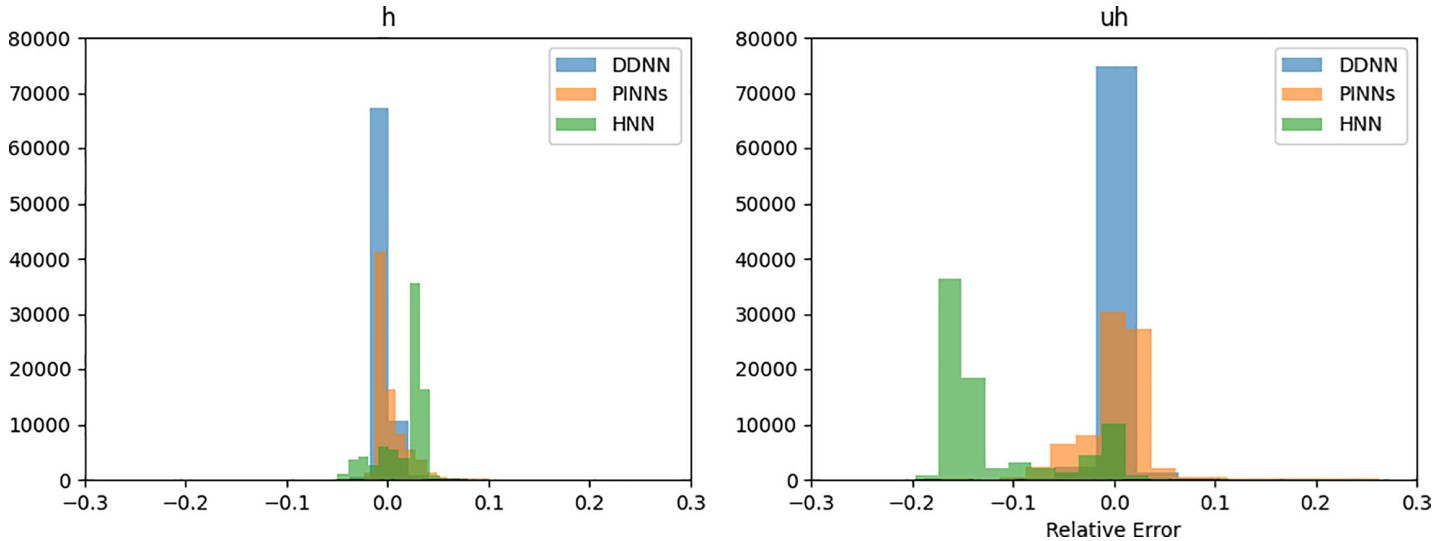

**Fig 11. Relative error distribution for *h*(*x*,*t*) and *uh*(*x*,*t*) for PINN, HNN and DDNN estimates.**

integrates both numerical data and PDE residuals, it uses only 1000 points (roughly 1.25%) from the available 80200 numerical data points for training. This limited data input may be insufficient to robustly guide the model toward accurate solutions.

In contrast, the PINN achieved better convergence and accuracy using just 400 points, indicating that physics-driven constraints alone can be efficient. The simultaneous incorporation of both data and physical constraints in the HNN may have introduced additional complexity in the training latent space, making optimization more challenging. Increasing the quantity of training data in the hybrid model could help it better integrate the complementary strengths of both data-driven and physics-informed learning.

To gain insight into the differences among the neural network estimates, we analyzed the contributions of individual components to the total loss function- see Fig 12. As expected, the DDNN relies solely on data mismatch losses, and there is a smooth convergence of the loss function with an accuracy of $10^{-6}$. For PINN, the loss comprises the PDE residual loss and initial condition (IC) loss. It can be observed that the PDE loss initially starts high but approaches $10^{-6}$ accuracy, whereas the IC loss converges at a lower accuracy of $10^{-4}$. The IC residual clearly dominates the loss landscape, where the network struggles to minimize the residuals effectively. In contrast, the HNN, which incorporates both data and physics constraints, exhibits a more balanced contribution between data fidelity and physical consistency. However, this balance is sensitive to the relative weighting of the individual loss terms. While the HNN benefits from improved stability for the convergence of individual loss terms compared to PINN, its overall accuracy is lower, which explains the higher error levels observed in its predictions.

Having observed the strengths and limitations of different neural network architectures in the 1D setting, now extend the analysis to more complex 2D dam break scenarios using standard PINN.

## 4.2 2D dam-break

The estimates of physics-informed neural networks for the 2D SWE applied to dam break scenarios described in Sect 2 are presented here. Each scenario is modeled using a separate PINN, training to approximate the solution over space and time. For training, a total of 25000 collocation points are uniformly sampled from the interior of the domain. To accurately capture the initial state, 25000 points are also placed in the interior domain at the initial time. The

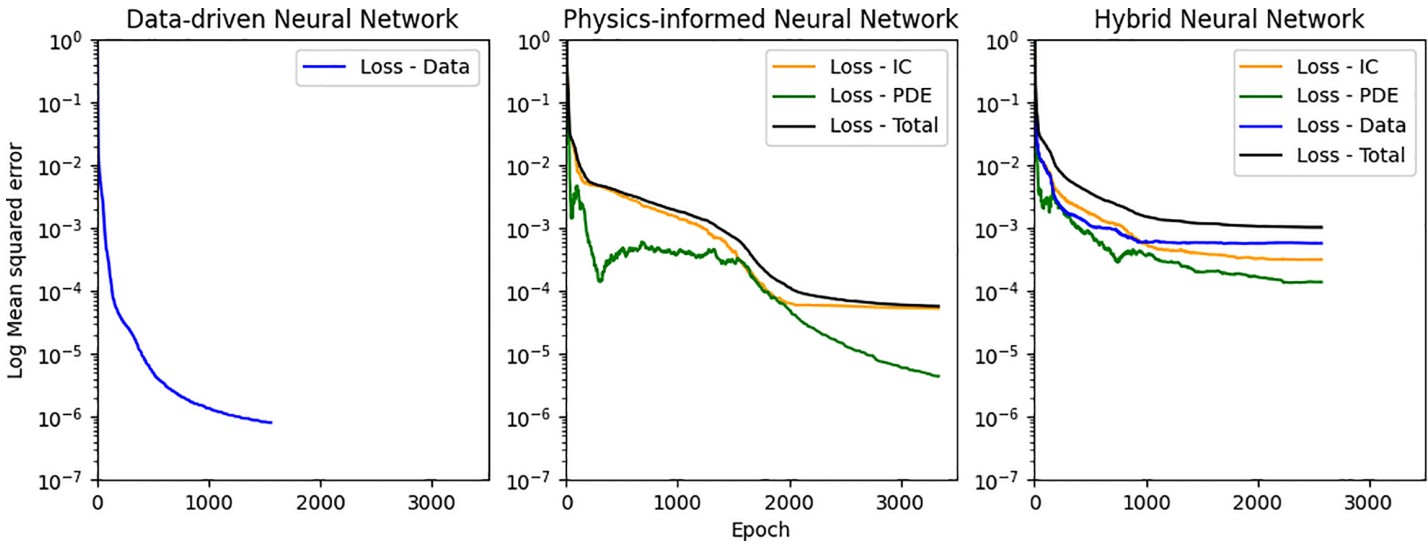

**Fig 12. Mean Squared errors vs Epoch for individual components of DDNN (left), PINN (middle), and HNN (right) loss function for 1-D Dam break use case.**

network architecture used in all cases consists of five hidden layers with 100 neurons each, using the hyperbolic tangent (tanh) activation function and Glorot uniform weight initialization. Training is performed using the Adam optimizer with a fixed learning rate of $10^{-3}$ running for a maximum $2 \times 10^4$ iterations. The code implementation is developed in Python 3, using DeepXDE as the primary framework for solving the governing equations. Built on TensorFlow, DeepXDE enables automatic differentiation and efficient training of deep learning models. All simulations are conducted in Google Colab, leveraging GPU acceleration for improved computational performance. Specifically, the models are trained on an NVIDIA Tesla T4 GPU, significantly reducing training time compared to CPU-based environments. Fig 13 shows the convergence behavior of the loss functions over training epochs.

Among the dam-break scenarios, Variant 4 exhibits the most stable convergence, with residuals in the loss term approaching zero and a mean squared error (MSE) on the order of $10^{-2}$. Variant 2 follows, reaching an MSE of approximately $10^{-1}$, indicating a moderately accurate approximation relative to the Gaussian profile. In contrast, Variant 3 (circular dam structure) and Variant 1 (stepped dam structure) display less favorable convergence trends. Figs 14–17 compare neural network predictions with reference simulations. Over-smoothed estimates are observed near sharp gradients, indicating persistent deviations in the predicted solutions that may explain the slower convergence during training.

Fig 18 shows the absolute error distributions for the four variants. Variant 4 (Gaussian) exhibits the tightest concentration around zero with a very short tail, while Variant 2 (Rectangular) is moderately spread. Variant 1 (Stepped) displays a pronounced right tail, with occasional large errors near the discontinuity, and Variant 3 (Circular) has the widest spread

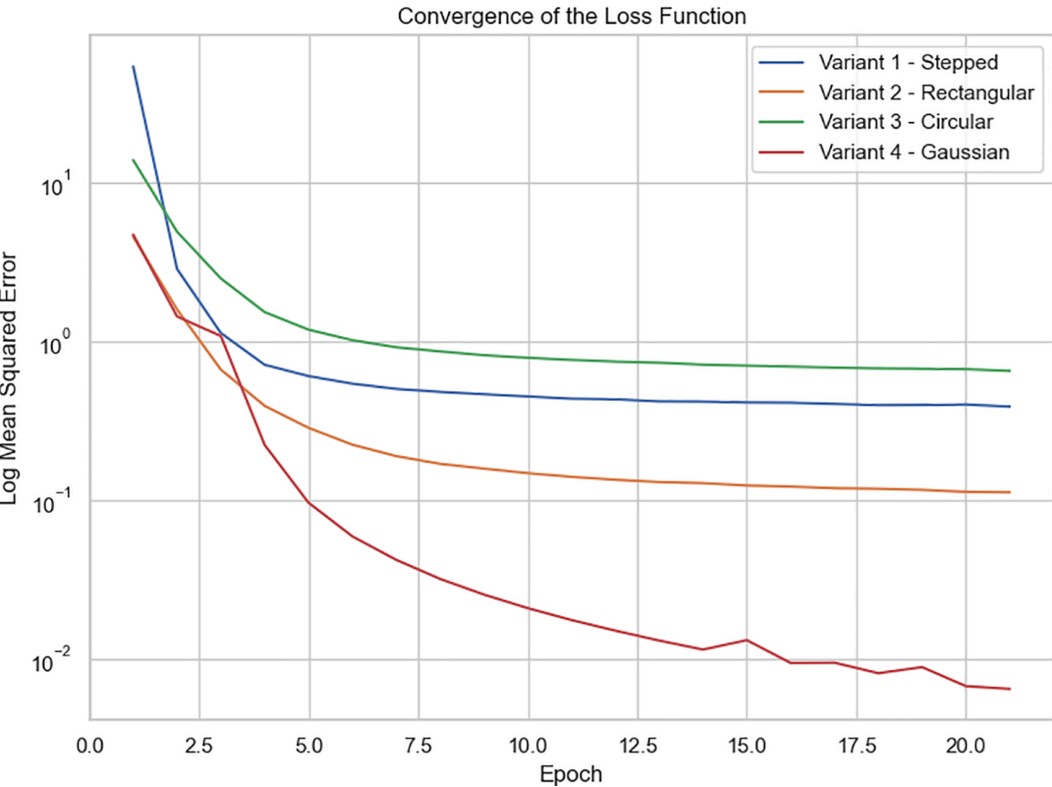

**Fig 13. Convergence of the mean squared error for the 2D variants 1-4.**

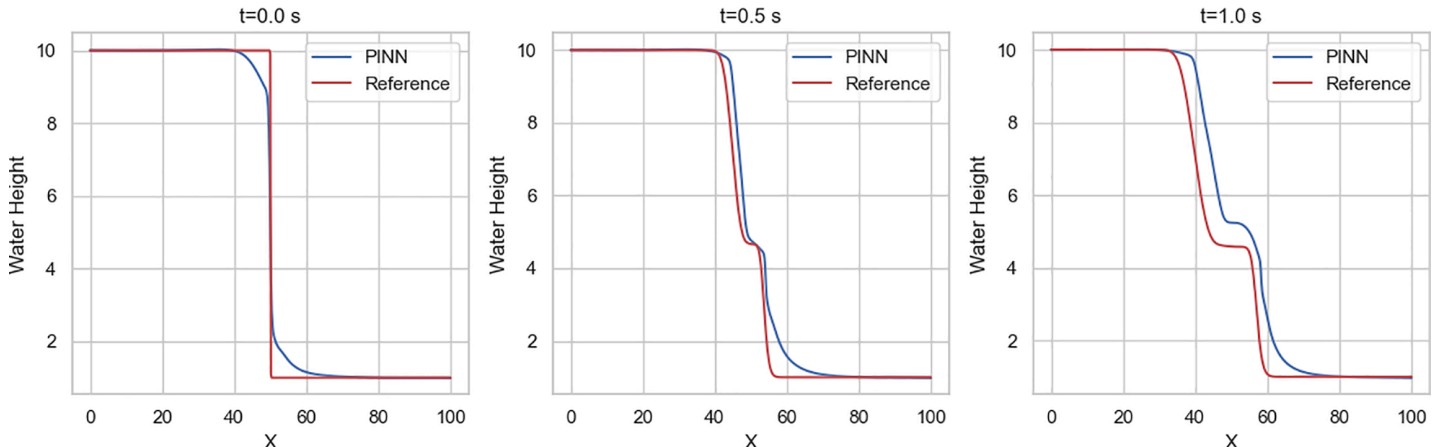

**Fig 14. Variant 1: Comparing the estimates from PINN against reference (Lax-Wendroff) solutions for water height $h(x)$ at center line $y = 50$ for time $t = 0, 0.5$ and 1 s.**

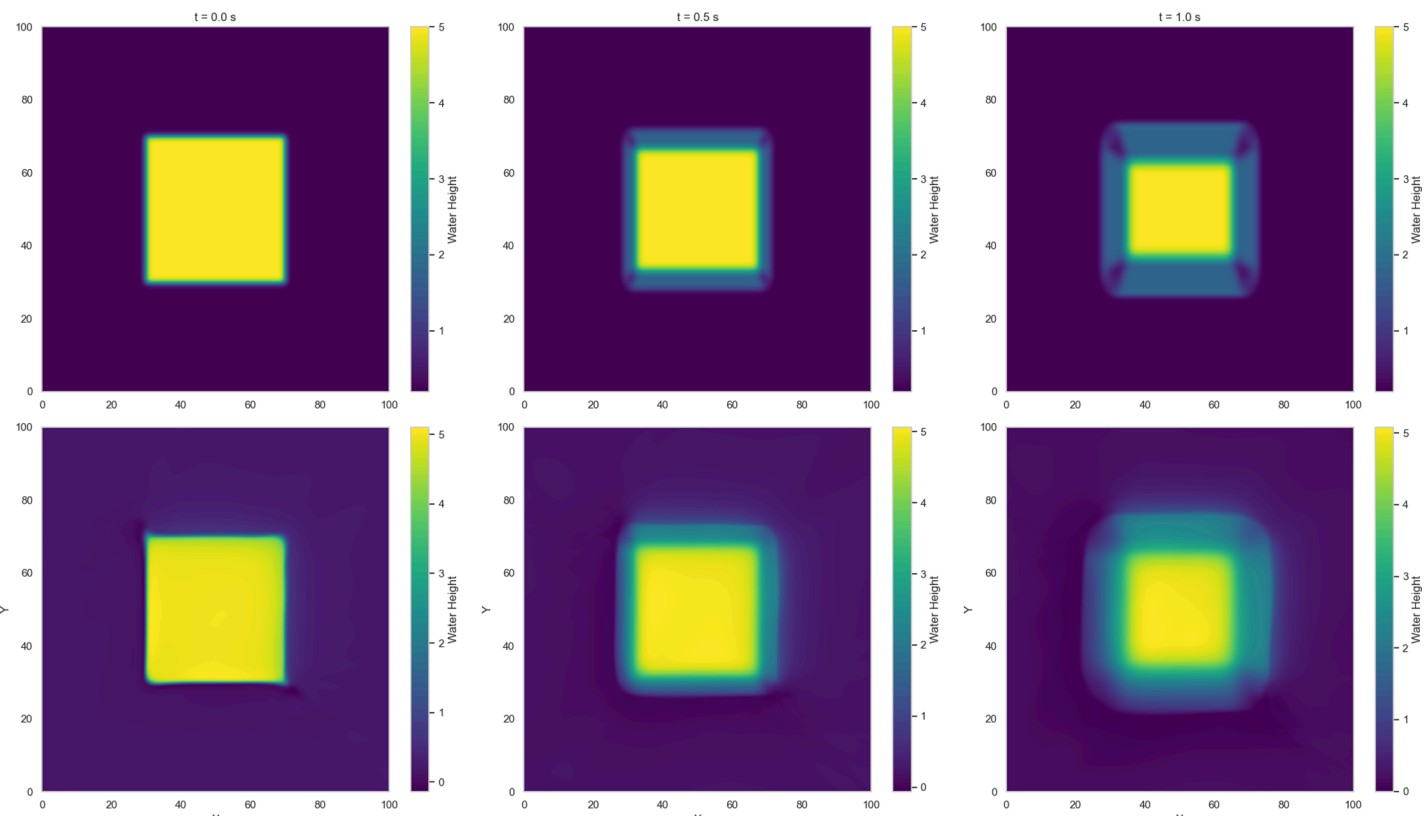

**Fig 15. Variant 2: Comparing the reference (Lax-Wendroff) solutions (above) and estimates from PINN (below) for water height $h(x)$ at center line $y = 50$ for time $t = 0, 0.5$ and 1 s.**

with the largest outliers. These results suggest that smoother initial profiles produce lower errors, whereas sharp fronts and curved wave interactions remain challenging, leading to localized high-error regions.

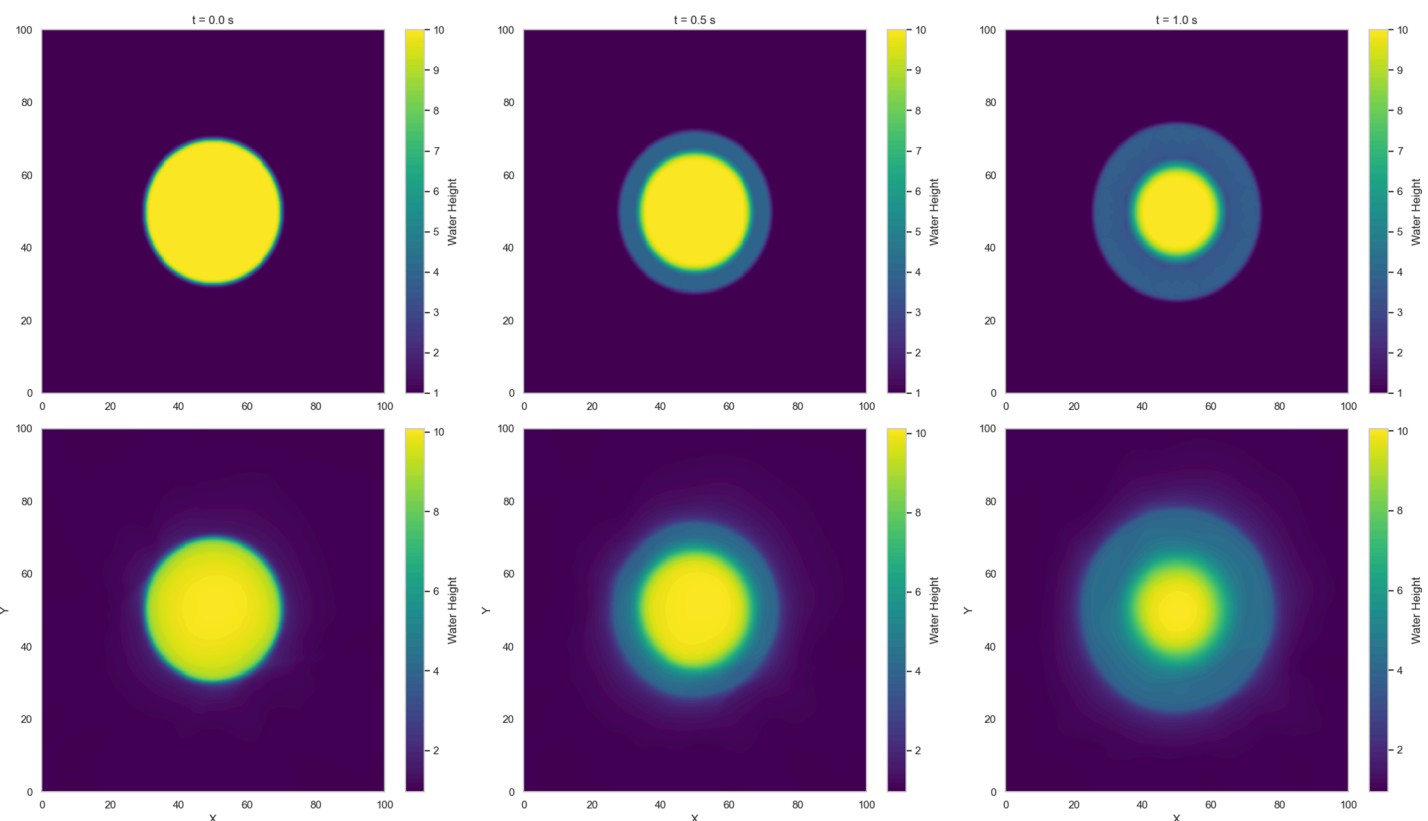

**Fig 16. Variant 3: Comparing the reference (Lax-Wendroff) solutions (above) and estimates from PINN (below) for water height $h(x)$ at center line $y = 50$ for time $t = 0$, 0.5 and 1 s.**

### Hybrid Neural Network (HNN) using precursor-guided refinement

In Fig 16, the first approach - relying solely on PINN from the outset - offers a straightforward implementation but struggles to accurately capture discontinuities, resulting in less reliable predictions in regions with complex flow dynamics around steep gradients. In contrast, the hybrid approach uses numerical solutions for the time interval [0,0.7] s as additional data-constraints alongside physics-based constraints to train the network.

The results from the hybridized PINN (HNN) in Fig 19 show that incorporating precursor numerical solutions for $t \in [0, 0.7]$ improves the network's accuracy for the circular dam-break scenario, especially in regions with steep gradients. This further highlights the advantage of precursor-guided training in capturing complex flow features while preserving physical fidelity.

Moreover, Fig 20 (Left) shows that incorporating precursor numerical simulations enhances the convergence behavior of the hybrid model. The log mean squared error for the HNN decreases more steadily and yields more accurate estimates over time. Similar convergence patterns were observed in the rectangular and Gaussian dam-break cases, where the loss stabilized around $10^{-3}$, indicating stronger adherence to physics-based constraints and more consistent learning across varying initial height profiles, particularly those with steep gradients. While this trend suggests a better-trained network, it does not by itself conclusively demonstrate improved generalization or robustness beyond the tested scenarios. Fig 20 (Right) further shows that the HNN produces lower errors compared to the PINN estimates.

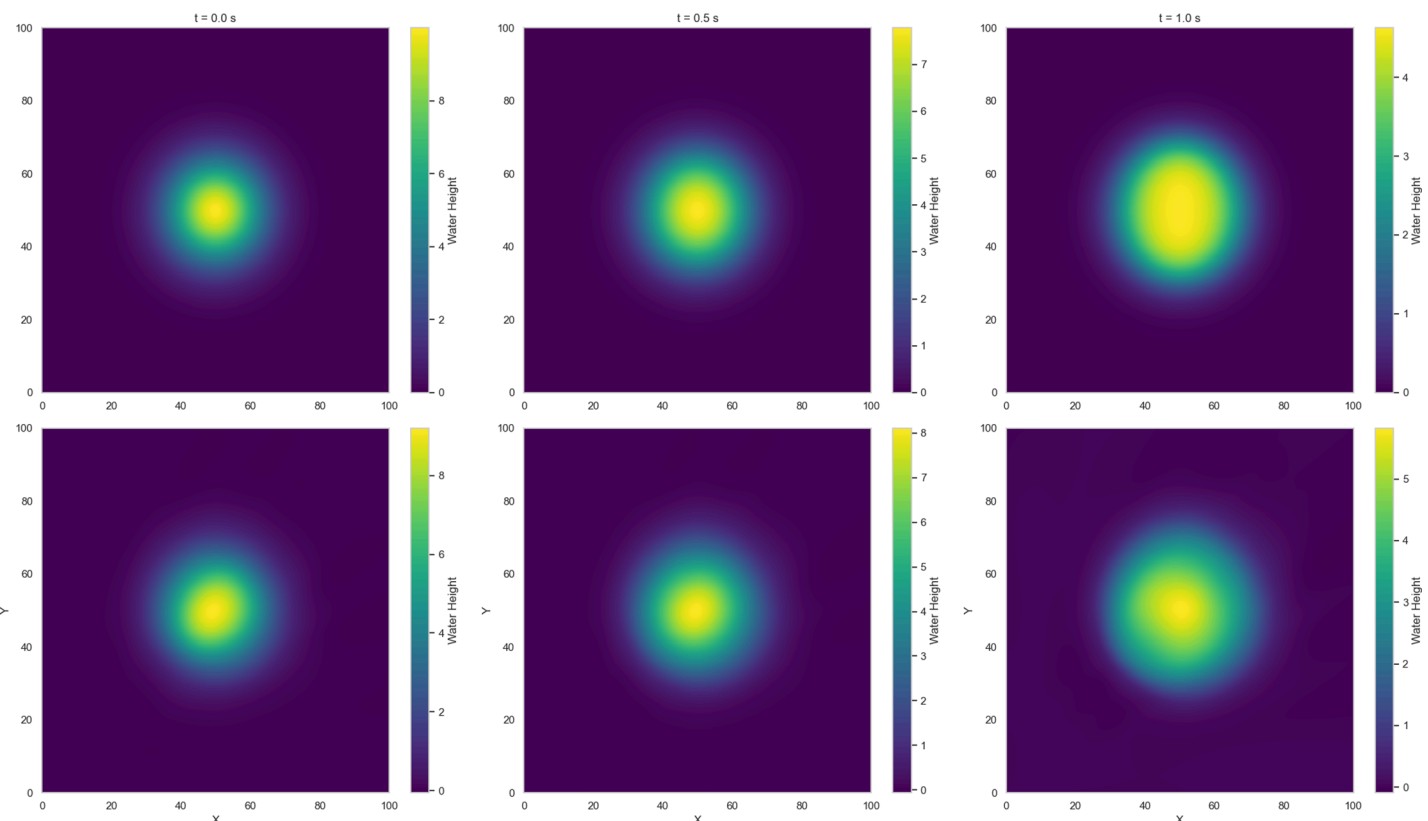

**Fig 17. Variant 4: Comparing the reference (Lax-Wendroff) solutions (above) and estimates from PINN (below) for water height *h(x)* at center line *y* = 50 for time *t* = 0, 0.5 and 1 s.**

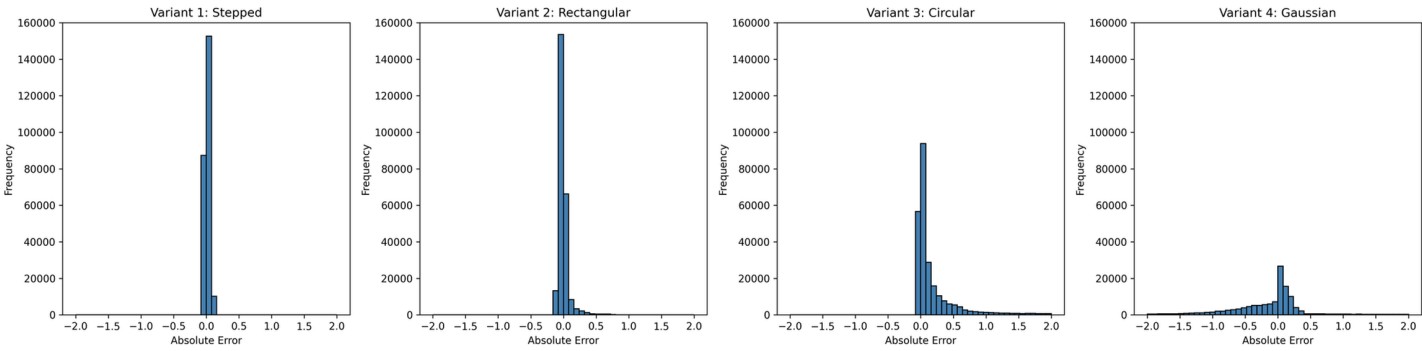

**Fig 18. Absolute error distribution for Variant 1, 2,3, and 4 at time *t* = 0.5 s.**

Table 1 summarizes the training time taken and accuracy achieved for the networks discussed above. While network training can be more computationally intensive than traditional methods, the flexibility of neural networks makes them well-suited for modeling complex, nonlinear dynamics. When trained with sufficient topographic detail, they show strong potential for delivering accurate, real-time estimates in dam-break scenarios.

Nevertheless, discontinuous initial profiles remain a significant challenge, requiring careful tuning of hyperparameters to achieve well-trained models. In particular, ensuring

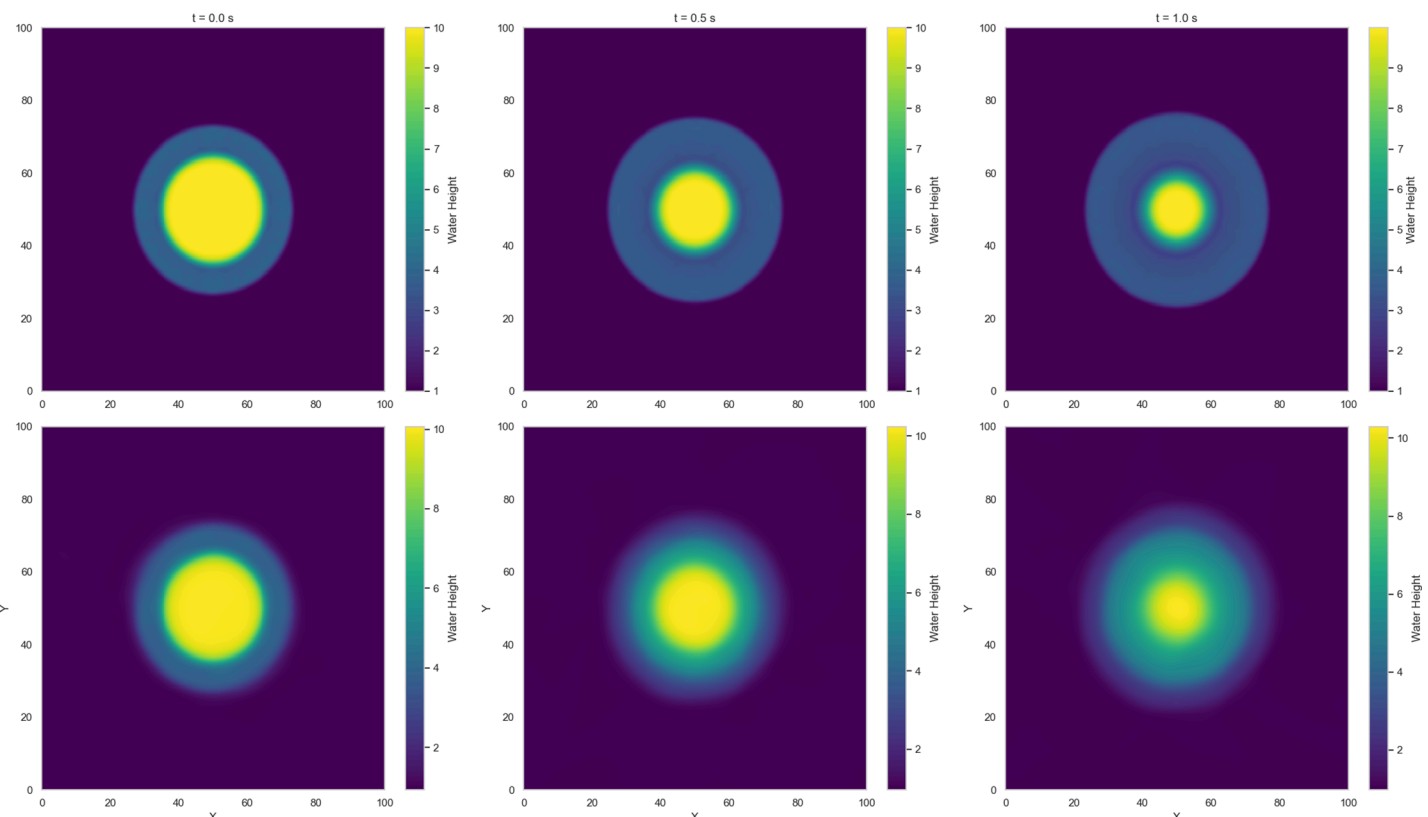

**Fig 19. Variant 3: Comparing the estimates from HNN against reference (Lax-Wendroff) solutions for water height $h(x)$ at center line $y = 50$ for time $t = 0, 0.5$ and 1 s.**

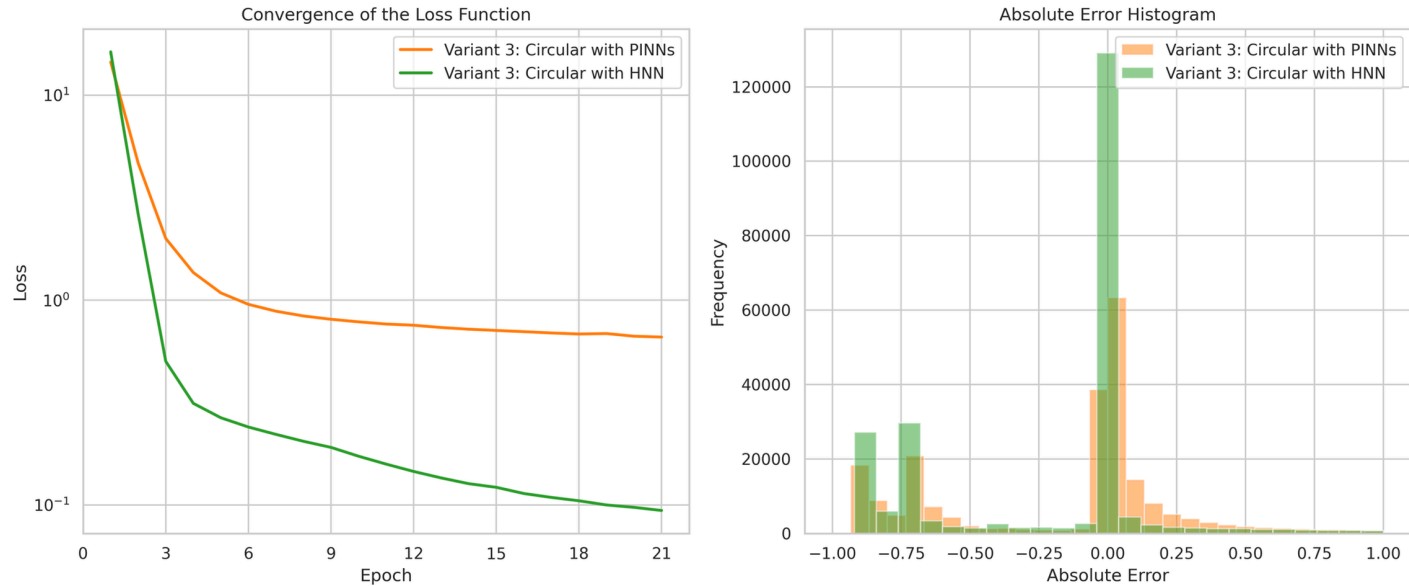

**Fig 20. Results for Variant 3 - circular dam structure.** Left: Convergence of the loss function for PINN and HNN. Right: Absolute error distribution at time $t = 0.5$ s for PINN and HNN

**Table 1. Performance statistics for training networks on NVIDIA's Tesla T4 GPU.**

| Height profile | Neural net | Training time (s) | Loss error |
|---|---|---|---|
| Step | PINN | 2848.7 | $4.01 \times 10^{-1}$ |
| Rectangular | PINN | 2504.2 | $2.67 \times 10^{-1}$ |
| Circular | PINN | 3018.5 | $6.60 \times 10^{-1}$ |
| Circular (improved) | HNN | 3131.1 | $9.40 \times 10^{-2}$ |
| Gaussian | PINN | 2840.8 | $9.08 \times 10^{-3}$ |

that all components of the loss function converge to zero or near-zero residuals demands additional effort. The inclusion of precursor simulations highlights the promise of hybrid strategies that combine numerical solutions with physics-based constraints, offering a viable pathway to improve both the training efficiency and reliability of neural networks in complex flow scenarios.

## 5 Conclusion

Similar to the work of [28], our study demonstrates that PINN holds potential as an efficient tool for providing real-time estimates of dam-break scenarios governed by the shallow water equation. However, sharp discontinuities in the initial conditions remain a key challenge, necessitating further investigation into hyperparameter tuning and adaptive training strategies to ensure convergence across all components of the loss function.

Compared to traditional numerical schemes such as Lax-Wendroff, the neural network-based approach results in smoother estimates, especially in regions with steep gradients. Our experiments with 1D and 2D dam-break scenarios confirm that PINN often struggles in these regions - a limitation also discussed in depth by [29]. To address this, we introduced a hybrid strategy that incorporates precursor numerical simulations, building upon ideas similar to those proposed in [30]. Our approach focused on two-dimensional dam-break problems, including both rectangular and circular dam profiles. In the circular dam-break case, we found that providing numerical reference data during the early time window improves the quality of the trained network. This hybrid strategy helps mitigate the issue arising from discontinuous initial profiles, which are common in hydrodynamic systems with complex terrain or boundary conditions. These findings are also in line with [16], which emphasizes the impact of gradient pathologies and training stiffness on PINN performance.

While PINN can be more computationally demanding than traditional numerical methods - especially when dealing with complex, nonlinear behaviors - its flexibility offers notable advantages. If trained effectively, particularly with the incorporation of complex topography profiles, PINN has the potential to deliver accurate and fast real-time estimates for dam-break scenarios. This makes them a promising tool for data-driven flood prediction and rapid decision support in hydrodynamic modeling.

Looking ahead, we believe that the performance of PINN can be further improved by incorporating additional data constraints into the training process. Further research should focus on extending the current framework to accommodate complex topography profiles and precursor simulations, in conjunction with collocation-based training and the governing physical equations. A key direction will be to apply the framework to more general flow scenarios governed by the unsteady Navier–Stokes equations. On the network design side, further exploration of adaptive activation functions, sensitivity analyses on layer-node configurations, and targeted regularization techniques will be essential. Finally, reducing computational overhead and advancing automated hyperparameter optimization will be

critical steps toward making PINN a scalable and practical solution for real-world engineering applications.

## Supporting information

**S1 File.** Dataset for the 1D and 2D shallow water equations (SWE). Repository: https://github.com/kinzamumtaz/2D_SWE.
(ZIP)

## Author contributions

**Conceptualization:** Zahra Lakdawala.

**Formal analysis:** Muhammad Waasif Nadeem.

**Investigation:** Kinza Mumtaz.

**Methodology:** Kinza Mumtaz.

**Resources:** Kinza Mumtaz.

**Software:** Kinza Mumtaz.

**Supervision:** Adnan Khan.

**Validation:** Kinza Mumtaz, Muhammad Waasif Nadeem.

**Visualization:** Kinza Mumtaz.

**Writing – original draft:** Kinza Mumtaz.

**Writing – review & editing:** Kinza Mumtaz, Zahra Lakdawala.

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
