## [Decision Letter · Decision Letter 0]

6 Jun 2025

PONE-D-25-12306HYDRODYNAMIC MODELING USING PHYSICS INFORMED MACHINE LEARNING METHODPLOS ONE

Dear Dr. Mumtaz,

Thank you for submitting your manuscript to PLOS ONE. After careful consideration, we feel that it has merit but does not fully meet PLOS ONE’s publication criteria as it currently stands. Therefore, we invite you to submit a revised version of the manuscript that addresses the points raised during the review process.

We look forward to receiving your revised manuscript.

Kind regards,

Divyesh Varade, PhD

Academic Editor

PLOS ONE

**Journal Requirements:**

1. When submitting your revision, we need you to address these additional requirements. Please ensure that your manuscript meets PLOS ONE's style requirements, including those for file naming. The PLOS ONE style templates can be found at https://journals.plos.org/plosone/s/file?id=wjVg/PLOSOne_formatting_sample_main_body.pdf and https://journals.plos.org/plosone/s/file?id=ba62/PLOSOne_formatting_sample_title_authors_affiliations.pdf 2. Please update your submission to use the PLOS LaTeX template. The template and more information on our requirements for LaTeX submissions can be found at http://journals.plos.org/plosone/s/latex. 3. Please note that PLOS ONE has specific guidelines on code sharing for submissions in which author-generated code underpins the findings in the manuscript. In these cases, we expect all author-generated code to be made available without restrictions upon publication of the work. Please review our guidelines at https://journals.plos.org/plosone/s/materials-and-software-sharing#loc-sharing-code and ensure that your code is shared in a way that follows best practice and facilitates reproducibility and reuse. 4. We note that your Data Availability Statement is currently as follows: All relevant data are within the manuscript and its Supporting Information files. Please confirm at this time whether or not your submission contains all raw data required to replicate the results of your study. Authors must share the “minimal data set” for their submission. PLOS defines the minimal data set to consist of the data required to replicate all study findings reported in the article, as well as related metadata and methods (https://journals.plos.org/plosone/s/data-availability#loc-minimal-data-set-definition). For example, authors should submit the following data: - The values behind the means, standard deviations and other measures reported;- The values used to build graphs;- The points extracted from images for analysis. Authors do not need to submit their entire data set if only a portion of the data was used in the reported study. If your submission does not contain these data, please either upload them as Supporting Information files or deposit them to a stable, public repository and provide us with the relevant URLs, DOIs, or accession numbers. For a list of recommended repositories, please see https://journals.plos.org/plosone/s/recommended-repositories. If there are ethical or legal restrictions on sharing a de-identified data set, please explain them in detail (e.g., data contain potentially sensitive information, data are owned by a third-party organization, etc.) and who has imposed them (e.g., an ethics committee). Please also provide contact information for a data access committee, ethics committee, or other institutional body to which data requests may be sent. If data are owned by a third party, please indicate how others may request data access. 5. PLOS requires an ORCID iD for the corresponding author in Editorial Manager on papers submitted after December 6th, 2016. Please ensure that you have an ORCID iD and that it is validated in Editorial Manager. To do this, go to ‘Update my Information’ (in the upper left-hand corner of the main menu), and click on the Fetch/Validate link next to the ORCID field. This will take you to the ORCID site and allow you to create a new iD or authenticate a pre-existing iD in Editorial Manager. 6. Please amend either the abstract on the online submission form (via Edit Submission) or the abstract in the manuscript so that they are identical.

**Additional Editor Comments:**

I have received the reviewer reports and based on these and my own evaluation of the manuscript, I invite the authors to revise the manuscript in accordance with the comments.

Reviewers' comments:

Reviewer's Responses to Questions

**Comments to the Author**

1. Is the manuscript technically sound, and do the data support the conclusions?

Reviewer #1: Yes

Reviewer #2: Partly

Reviewer #3: Partly

2. Has the statistical analysis been performed appropriately and rigorously? 

Reviewer #1: Yes

Reviewer #2: N/A

Reviewer #3: Yes

3. Have the authors made all data underlying the findings in their manuscript fully available?

Reviewer #1: Yes

Reviewer #2: No

Reviewer #3: No

4. Is the manuscript presented in an intelligible fashion and written in standard English?

Reviewer #1: Yes

Reviewer #2: Yes

Reviewer #3: Yes

5. Review Comments to the Author

**Reviewer #1:** I enjoy reading the paper. Some specific comments are as follow:

1. What is meant by: This approach can improve the accuracy of predictions, especially when data are limited, noisy, or missing.

2. The training setup is described in 3.0.2. Is the additional description in 4.0 redundant or for a different purpose?

3. How are the number of layers and neuron decided a priori?

4. The distinction between the PINN, HNN and DDNN is not entirely clear for me.

(a) For PINN, what is the training input?

(b) For HNN, what is the training input? What is done in t ∈ (0, 0.4) and t ∈ (0.4, 1) for each/both the approach?

(c) For DDNN, what is the training input?

Perhaps a graphical summary/ flow chart will greatly help the reader.

Note: Meaning of the term "unseen data" is not clear. Care should be taken to differentiate "data", "input", "numerical results (data?)"

5. Please provide detail on: the loss function formulation differs for each of the networks.

6. Statistical justification for: outperforming traditional numerical methods in terms of generalization and computational efficiency.

7. Please clarify statement in Abstract: This study emphasizes PINNs scalability and versatility for hydrodynamic modeling,

Other area to improve:

1. Paragraph 1 is too lengthy and should be split accordingly.

2. The figures quality is not adequate for viewing the results in detail and the legend

**Reviewer #2: **1. In the introduction section, the sentence beginning with "In recent years, … of hydrological applications" lacks clarity. Please rephrase it to convey a more precise meaning.

2. The manuscript currently cites only a few references ([1], [2], [3]) and lacks an in-depth comparison of related work involving PINNs in hydrology or fluid mechanics. A more comprehensive literature review is needed to establish the novelty and context of your work.

3. The manuscript contains several repetitive sentences, especially in the introduction and methodology sections. Consider revising for conciseness and clarity.

4. In Section 1.1 (State of the Art), phrases like "[4] is employed..." should be rewritten using author names, e.g., "Kader et al., 2020 [4]." Please ensure that all references throughout the manuscript follow this format for clarity and readability.

5. Figures 7, 9, 11, 13, and 15 are labeled as showing both training and testing loss, but only one curve is plotted, which appears to be the training loss. Moreover, these figures are not cited or discussed in the main text. Please correct the figures and provide a proper interpretation in the results section.

6. The purpose and content of Figures 4 and 6 are unclear. Please elaborate on what these figures represent and how they contribute to the findings.

7. The manuscript does not explain how the models were trained. What portions of the data were used for training and testing? What were the input features and target variables? This information is critical and must be included.

8. There is no discussion of how model performance was validated. Please include quantitative evaluation metrics such as RMSE, MAE, MSE, or R² to assess and compare the model predictions.

9. The results section lacks detailed explanation and interpretation of the figures. Each figure should be properly described and analyzed to highlight key insights and outcomes.

10. A discussion section is completely missing from the manuscript. Please add a dedicated section that critically interprets the findings, compares them with previous studies, discusses limitations, and highlights the implications of your results.

**Reviewer #3:** This study proposed a Physics-Informed Neural Network (PINN) framework to solve 1D and 2D SWEs for dam-break problems. While some of their results are interesting, the study has low novelty regarding methods and applications. Some papers, such as 10.1016/j.jhydrol.2024.131263, 10.1029/2023WR036589, 10.48550/arXiv.2406.16236, etc., all trained the PINN without precomputed numerical data.

The main comments are:

1. The paper is only an application of PINN with no new idea or approach. The contribution of this paper is unclear.

2. The test cases are very simple. No friction or varying topography is considered. Besides, I do not appreciate that the wet-dry problem is ignored in the study. I would recommend the paper to consider at least one of these issues.

3. The comparison between PINN and other numerical methods is unfair. In this study, PINN is compared with a FD scheme, which is well known to struggle with shock wave modelling and has been considered suboptimal for such water problems for over a decade. To strengthen the evaluation, I recommend comparing PINN with more advanced numerical methods, such as FV and FE schemes, which are more robust in handling discontinuities and complex flow dynamics.

Other comments are:

1. Introduction: Shallow water modelling belongs to the hydraulic study instead of the hydrological study. More references for flood review should be cited.

2. The review of using PINN to solve SWEs is limited. The study of PINN for SWEs is recently a very active field. Many papers can be cited. More examples such as doi.org/10.3389/fcpxs.2024.1508091, 10.1029/2024WR037490, 10.1016/j.scitotenv.2023.168814, 1016/j.jcp.2022.111024.

3. The paper could include a table summarising all the PINN training configurations.

4. Some experiments for the model design and trials can be added as the appendix.

6. PLOS authors have the option to publish the peer review history of their article (what does this mean?). If published, this will include your full peer review and any attached files.

Reviewer #1: No

Reviewer #2: No

Reviewer #3: **Yes: **Xin Qi

---

## [Author Response · Author response to Decision Letter 1]

21 Aug 2025

We would like to express our sincere gratitude to the reviewers for their valuable comments and suggestions, which have greatly helped to improve the manuscript. Their insights have been instrumental in refining the work. In light of the comments, we have updated the title and overall structure of the paper. We now give a pointwise response to the reviewer’s comments.

Reviewer #1: I enjoy reading the paper. Some specific comments are as follow:

1. What is meant by: This approach can improve the accuracy of predictions, especially when data are limited, noisy, or missing.

Answer: This refers to the hybrid PINNs framework using physics and limited data.

We added a clarification in Section 1 (Introduction) with references (e.g., Raissi et al. 2019, Mao et al. 2020) to support this claim.

2. The training setup is described in 3.0.2. Is the additional description in 4.0 redundant or for a different purpose?

Answer: To avoid redundancy, we streamlined and updated Section 4.2 for shared parameters and retained case-specific details.

3. How are the number of layers and neuron decided a priori?

Answer: We added an explanation and justification in Section 3.0 with references.

4. The distinction between the PINN, HNN and DDNN is not entirely clear for me.

(a) For PINN, what is the training input?

Answer: This has been made clearer in the manuscript, for example, see section 3.

(b) For HNN, what is the training input? What is done in t ∈ (0, 0.4) and t ∈ (0.4, 1) for each/both the approach?

Answer: Numerical data (t ∈ [0,0.4]) + Physics, for more details, see section 3.3. In the second time interval, there were no interesting dynamics; we have removed it from the manuscript.

(c) For DDNN, what is the training input?

Answer: Numerical data (t ∈ [0,0.4]) for more details, see section 3.1.

Perhaps a graphical summary/ flow chart will greatly help the reader.

Answer: This has been made clearer in the revised manuscript

Note: Meaning of the term "unseen data" is not clear. Care should be taken to differentiate "data", "input", "numerical results (data?)"

Answer: In the second time interval, where results were mentioned for “unseen data”, there were no interesting dynamics; we have removed it from the manuscript.

5. Please provide detail on: the loss function formulation differs for each of the networks.

Answer: Added in sections 3.1, 3.2, 3.3

6. Statistical justification for: outperforming traditional numerical methods in terms of generalization and computational efficiency.

Answer: A table has been added in Section 4.2.

7. Please clarify statement in Abstract: This study emphasizes PINNs scalability and versatility for hydrodynamic modeling,

Answer: The abstract has been updated with clearer wording.

Other area to improve:

1. Paragraph 1 is too lengthy and should be split accordingly.

Answer: The manuscript has been updated accordingly.

2. The figures quality is not adequate for viewing the results in detail and the legend

Answer: The manuscript has been updated accordingly.

Reviewer #2: 1. In the introduction section, the sentence beginning with "In recent years, … of hydrological applications" lacks clarity. Please rephrase it to convey a more precise meaning.

Answer: The manuscript has been updated accordingly.

2. The manuscript currently cites only a few references ([1], [2], [3]) and lacks an in-depth comparison of related work involving PINNs in hydrology or fluid mechanics. A more comprehensive literature review is needed to establish the novelty and context of your work.

Answer: More references have been added in the introduction.

3. The manuscript contains several repetitive sentences, especially in the introduction and methodology sections. Consider revising for conciseness and clarity.

Answer: The manuscript has been updated accordingly.

4. In Section 1.1 (State of the Art), phrases like "[4] is employed..." should be rewritten using author names, e.g., "Kader et al., 2020 [4]." Please ensure that all references throughout the manuscript follow this format for clarity and readability.

Answer: The reference format has been updated and made consistent throughout.

5. Figures 7, 9, 11, 13, and 15 are labeled as showing both training and testing loss, but only one curve is plotted, which appears to be the training loss. Moreover, these figures are not cited or discussed in the main text. Please correct the figures and provide a proper interpretation in the results section.

Answer: The loss plots have been updated for clarity.

6. The purpose and content of Figures 4 and 6 are unclear. Please elaborate on what these figures represent and how they contribute to the findings

Answer: Explanation has been added in the manuscript.

7. The manuscript does not explain how the models were trained. What portions of the data were used for training and testing? What were the input features and target variables? This information is critical and must be included.

Answer: We have updated the manuscript to explicitly include this breakdown and added clarification regarding input-output pairs, training intervals, and evaluation domains, e.g, section 2.3, 3.0.

8. There is no discussion of how model performance was validated. Please include quantitative evaluation metrics such as RMSE, MAE, MSE, or R² to assess and compare the model predictions.

Answer: Added in section 4.2.

9. The results section lacks detailed explanation and interpretation of the figures. Each figure should be properly described and analyzed to highlight key insights and outcomes.

Answer: The manuscript has been updated accordingly.

10. A discussion section is completely missing from the manuscript. Please add a dedicated section that critically interprets the findings, compares them with previous studies, discusses limitations, and highlights the implications of your results.

Answer: The manuscript's structure and write-up have been updated to present our findings more clearly.

Reviewer #3: This study proposed a Physics-Informed Neural Network (PINN) framework to solve 1D and 2D SWEs for dam-break problems. While some of their results are interesting, the study has low novelty regarding methods and applications. Some papers, such as 10.1016/j.jhydrol.2024.131263, 10.1029/2023WR036589, 10.48550/arXiv.2406.16236, etc., all trained the PINN without precomputed numerical data.

The main comments are:

1. The paper is only an application of PINN with no new idea or approach. The contribution of this paper is unclear.

Answer: We appreciate the reviewer’s comment regarding the novelty and complexity of our work. While we agree that this study applies an existing PINNs framework rather than introducing a new algorithm, its core contribution lies in evaluating and comparing PINNs, HNNs, and DDNNs in the context of dam-break problems governed by the shallow water equations. Our study provides a systematic assessment of how different learning strategies—physics-based, hybrid, and data-driven—perform under identical test conditions.

2. The test cases are very simple. No friction or varying topography is considered. Besides, I do not appreciate that the wet-dry problem is ignored in the study. I would recommend the paper to consider at least one of these issues.

Answer: Regarding the simplicity of the test cases, this was a deliberate choice to isolate model behavior and stability in a controlled setting. We focused on evaluating baseline performance without the added complexity of frictional losses or wet-dry boundaries, which can obscure the learning dynamics and error sources in early-stage studies.

That said, we fully agree that modeling frictional effects, varying topography, or wetting-drying interfaces is essential for real-world flood prediction. We have now added a note in the conclusion section highlighting these as important directions for future work. We also acknowledge that wet-dry problems pose unique challenges for PINNs, especially in maintaining stability near dry interfaces, and addressing this would require specialized boundary treatment or adaptive sampling—both of which we are actively investigating.

3. The comparison between PINN and other numerical methods is unfair. In this study, PINN is compared with a FD scheme, which is well known to struggle with shock wave modelling and has been considered suboptimal for such water problems for over a decade. To strengthen the evaluation, I recommend comparing PINN with more advanced numerical methods, such as FV and FE schemes, which are more robust in handling discontinuities and complex flow dynamics.

Answer: We are just using a numerical scheme as a benchmark, and the goal is not to show the superiority and better performance of PINN over a particular numerical scheme. The study aims to validate and consider the performance of PINN, HNN, and DDNN.

Other comments are:

1. Introduction: Shallow water modelling belongs to the hydraulic study instead of the hydrological study. More references for flood review should be cited.

Answer: The manuscript has been updated accordingly.

2. The review of using PINN to solve SWEs is limited. The study of PINN for SWEs is recently a very active field. Many papers can be cited. More examples such as doi.org/10.3389/fcpxs.2024.1508091, 10.1029/2024WR037490, 10.1016/j.scitotenv.2023.168814, 1016/j.jcp.2022.111024.

Answer: More papers have been cited.

3. The paper could include a table summarising all the PINN training configurations.

Answer: The table has been added in section 4.2.

4. Some experiments for the model design and trials can be added as the appendix.

Answer: We have included further details concerning model design in the results section instead.

---

## [Decision Letter · Decision Letter 1]

3 Sep 2025

Investigating the use of physics informed neural networks for dam-break scenarios

PONE-D-25-12306R1

Dear Dr. Mumtaz,

We’re pleased to inform you that your manuscript has been judged scientifically suitable for publication and will be formally accepted for publication once it meets all outstanding technical requirements.

Kind regards,

Divyesh Varade, PhD

Academic Editor

PLOS ONE

Additional Editor Comments (optional):

The reviewers have recommended consideration of the revised manuscript for publication and I am in agreement with their comments. Reviewer-3 has mentioned including some review literature to be incorporated, which is optional.

Reviewers' comments:

Reviewer's Responses to Questions

**Comments to the Author**

1. If the authors have adequately addressed your comments raised in a previous round of review and you feel that this manuscript is now acceptable for publication, you may indicate that here to bypass the “Comments to the Author” section, enter your conflict of interest statement in the “Confidential to Editor” section, and submit your "Accept" recommendation.

Reviewer #1: All comments have been addressed

Reviewer #3: (No Response)

2. Is the manuscript technically sound, and do the data support the conclusions?

Reviewer #1: Yes

Reviewer #3: Yes

3. Has the statistical analysis been performed appropriately and rigorously? 

Reviewer #1: Yes

Reviewer #3: N/A

4. Have the authors made all data underlying the findings in their manuscript fully available?

Reviewer #1: Yes

Reviewer #3: Yes

5. Is the manuscript presented in an intelligible fashion and written in standard English?

Reviewer #1: Yes

Reviewer #3: Yes

6. Review Comments to the Author

Reviewer #1: (No Response)

Reviewer #3: I have reviewed all the responses and the whole paper. The authors may not want to do any extra tests. Thus, I will leave it to the editors to decide.

I would still recommend that authors have more references to support their discussion. Some studies have attempted to address issues, such as the wet-dry front or changing topography (https://doi.org/10.1016/j.jhydrol.2024.131263, https://doi.org/10.3390/w17081239, https://doi.org/10.3390/w17081239, https://doi.org/10.1016/j.ocemod.2025.102601 etc). I would encourage the authors to further improve the manuscript by enriching the reviewed papers.

7. PLOS authors have the option to publish the peer review history of their article (what does this mean?). If published, this will include your full peer review and any attached files.

Reviewer #1: No

Reviewer #3: No

---

## [Editor Report · Acceptance letter]

PONE-D-25-12306R1

PLOS ONE

Dear Dr. Mumtaz,

I'm pleased to inform you that your manuscript has been deemed suitable for publication in PLOS ONE. Congratulations! Your manuscript is now being handed over to our production team.

Kind regards,

on behalf of

Dr. Divyesh Varade

Academic Editor

PLOS ONE